# HARD VIEW SELECTION FOR SELF-SUPERVISED LEARNING

## ABSTRACT

Many Self-Supervised Learning (SSL) methods train their models to be invariant to different "views" of an image for which a good data augmentation pipeline is crucial. While considerable efforts were directed towards improving pre-text tasks, architectures, or robustness (e.g., Siamese networks or teacher-softmax centering), the majority of these methods remain strongly reliant on the random sampling of operations within the image augmentation pipeline, such as the random resized crop or color distortion operation. In this paper, we argue that the role of the view generation and its effect on performance has so far received insufficient attention. To address this, we propose an easy, learning-free, yet powerful Hard View Selection (HVS) strategy designed to extend the random view generation to expose the pretrained model to harder samples during SSL training. It encompasses the following iterative steps: 1) randomly sample multiple views and create pairs of two views, 2) run forward passes for each view pair on the currently trained model, 3) adversarially select the pair yielding the worst loss, and 4) run the backward pass with the selected pair. In our empirical analysis we show that under the hood, HVS increases task difficulty by controlling the Intersection over Union of views during pretraining. With only 300-epoch pretraining, HVS is able to closely rival the 800-epoch DINO baseline which remains very favorable even when factoring in the slowdown induced by the additional forwards of HVS. Additionally, HVS consistently achieves accuracy improvements on ImageNet up to 0.40% and 1.9% on linear evaluation and similar improvements on transfer tasks across multiple CL methods, such as DINO, SimSiam, iBOT and SimCLR.

## 1 INTRODUCTION

Various approaches to learn effective and generalizable visual representations in Self-Supervised Learning (SSL) exist. One way to categorize many SSL methods is to distinguish generative and discriminative approaches. While generative methods aim at generating image input, discriminative methods, with contrastive learning (Hadsell et al., 2006; He et al., 2020) being the most renowned approach, aim at learning a latent representation in which similar image views are closely and dissimilar ones are distantly located.

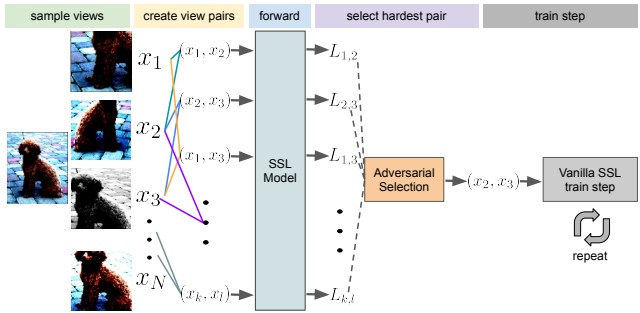

Figure 1: HVS first samples $N$ views, pairs them, and adversarially selects the hardest pair, i.e., the one with the worst loss according to the current model state.

Such views are generated by applying a sequence of (randomly sampled) image transformations and are usually composed of geometric (cropping, rotation, etc.) and appearance (color distortion, blurring, etc.) transformations. A body of literature (Chen et al., 2020a; Wu et al., 2020; Purushwalkam & Gupta, 2020; Wagner et al., 2022; Tian et al., 2020b) has illuminated the effects of image views on representation learning and identified *random resized crop* (RRC) transformation, which randomly crops the image and resizes it back to a fixed size, as well as color distortion as one of the

most essential ones for effective contrastive learning. However, despite this finding and to our best knowledge, little research has gone into identifying more effective ways for *selecting* or generating views to improve performance. In this paper, we seek to extend the random view generation to expose the pretrained model to harder samples during CL training.

Supervised Learning works that address this shortcoming in a distant manner and which attempt to leverage task difficulty include Spatial Transformer Networks (Jaderberg et al., 2015)and Adversarial AutoAugment (Zhang et al., 2020), which either directly learn geometric transformations or generate augmentation policies that yield hard samples for more robust feature learning. However, these approaches have not been used successfully for unsupervised learning and Adversarial AutoAugment is limited in leveraging the geometric perspective for view hardness control as it does not include RRC in its search space. Furthermore, and in contrast to what we propose, integrating any of these methods into existing contrastive learning pipelines is non-trivial.

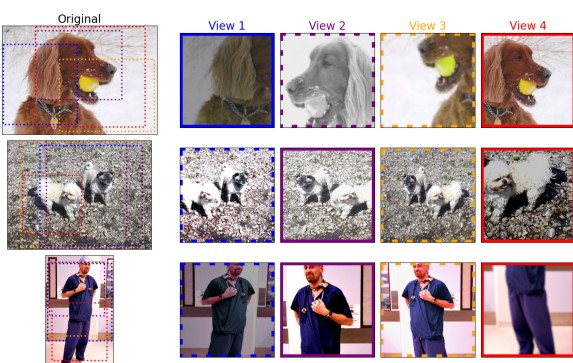

Figure 2: Examples (left) and sampled views (right) after resizing and appearance transformations. Pairs selected by HVS are shown with a solid frame.

Contrastive learning approaches that are related to our work and aim at controlling view hardness include (Shi et al., 2022) and (Tamkin et al., 2021) which propose masking or perturbing input images through adversarial models. Another approach is MADAug (Hou et al., 2023) which use a cooperative setting to learn policies yielding progressively harder samples based on a validation loss which is usually unavailable in contrastive learning. A work that takes a similar perspective like us in leveraging the information content of views for contrastive learning is (Tian et al., 2020b). Using this framework, they discover sweet spots for data augmentation hyperparameters that are widely used and adversarially learn flow-based models that generate novel color space-views which are used then for contrastive learning. While these works offer important insights into view optimality, they usually require learning additional models prior to the actual SSL training or non-trivial changes to the pipeline and thus, are not easily integrated into existing CL methods.

In contrast, we propose a fully learning-free and easy-to-integrate approach. Importantly, and different to Tian et al. (2020b), our approach leverages the current model state and operates on a per-sample basis during pretraining. It applies the following simple strategy: we sample a few views and forward each pair through the currently trained model. Next, we identify the pair that yields the worst loss among all pairs, use it for the backward pass, and repeat (see Fig. 1). While being agnostic to any contrastive loss, this strategy exposes the pretrained model to samples that the model at any given point in the training trajectory finds challenging. As our results show, sampling only four views are sufficient to achieve noticeable improvements.

Overall, our contributions can be summarized as follows:

- We propose *Hard View Selection (HVS)*, a new learning method complementing contrastive learning that is easy to integrate and extends the broadly-used random view generation to automatically expose the model to harder samples during pretraining. The only requirement of HVS is that it needs to be able to compute sample-wise losses;

- We demonstrate the effectiveness and compatibility of our approach using ImageNet (Deng et al., 2009) pretraining across popular and widely-adopted contrastive learning methods that cover a diverse range of objectives such as SimSiam (Chen & He, 2021), DINO (Caron et al., 2021), iBOT (Zhou et al., 2021) and SimCLR (Chen et al., 2020a);

- HVS improves all baselines noticeably compared to using the common view generation and we show this on a diverse set of transfer datasets and tasks, including linear evaluation, finetuning, object detection and segmentation;

- We present insights into the underlying mechanisms of HVS. One key observation is that harder samples are achieved by scheduling the Intersection over Union conditioned on the

current model state and image input. We also provide further results on the effects of changing the number of sampled views, color augmentation and studies on longer training.

We make our PyTorch (Paszke et al., 2019) code, models, and all used hyperparameters publicly available under `https://anonymous.4open.science/r/hard-view-selection/`.

## 2 RELATED WORK

### 2.1 CONTRASTIVE LEARNING IN SSL

The core idea behind the contrastive learning framework is to learn image representations by contrasting positive pairs (two views of the same image) against negative pairs (two views of different images) (Hadsell et al., 2006). To work well in practice and to prevent model collapse, contrastive learning methods often require a large number of negative samples (Wu et al., 2018; van den Oord et al., 2018; Chen et al., 2020a; He et al., 2020; Tian et al., 2020a; Chen et al., 2020b) stored in memory banks (Wu et al., 2018; He et al., 2020) or, for instance, in the case of SimCLR, implicitly in large batches (Chen et al., 2020a). Non-contrastive approaches, such as BYOL (Grill et al., 2020), SimSiam (Chen & He, 2021), DINO (Caron et al., 2021) and others (Zbontar et al., 2021; Caron et al., 2020; Ermolov et al., 2021), are able to only use positive pairs without causing model collapse but rely on other techniques, such as Siamese architectures, whitening of embeddings, clustering, maximizing the entropy of the embeddings, momentum encoders and more.

### 2.2 OPTIMIZING FOR HARD AUGMENTATIONS AND VIEWS IN SSL

Past work has shown that optimizing augmentation policies or views directly can be beneficial. Hence, the realm of learning task-specific augmentation policies based on data has seen quick development (Cubuk et al., 2019; Ho et al., 2019; Lin et al., 2019; Zhang et al., 2020; Hataya et al., 2020; Müller & Hutter, 2021). Most related to our work among these approaches is Adversarial AutoAugment (AAA) (Zhang et al., 2020) that proposes to generate augmentation policies yielding hard samples by learning with an adversarial supervised objective. In a similar, but cooperative setting, MADAug (Hou et al., 2023) learns policies yielding progressively harder samples based on a validation loss. In contrast to our work, however, AAA and MADAug do not include the random resize crop operation into their search spaces and thus, are limited in leveraging the geometric perspective to control view hardness.

Unlike learning hard augmentations, approaches exist that optimize directly on the pixel level to control view hardness. For instance, Shi et al. (2022) adversarially learn semantically meaningful masks for a pretrained network to inpaint. Another approach perturbs input images using an adversarial model (Tamkin et al., 2021). Tian et al. (2020b) leverage, like us, the view content for better performance. Under the lens of mutual information (MI), they unify several data augmentation CL methods and derive an MI-based objective to adversarially learn flow-based view generators for SSL. While these methods can complement our approach, they are less easily integrated into existing contrastive pipelines due to the need for adversarial network training. In contrast to our work, Tian et al. (2020b) does not use the current model state to influence learning hardness and while their studies incorporate distance metrics, they do not consider the Intersection over Union (IoU) metric which we extensively make use of. Additionally, we use our distance metric analysis to better understand the selection of HVS and to assess predictability of good or hard pairs.

## 3 METHOD

### 3.1 CONTRASTIVE LEARNING FRAMEWORK

In this section, we introduce our approach which is also depicted in Algorithm 1. Many different contrastive learning (He et al., 2020) objectives exist, each characterized by variations stemming from design choices, such as by the use of positive and negative samples or asymmetry in the encoder-projector network structure. For simplicity of exposure, we will introduce our approach based on the SimSiam objective Chen & He (2021), but we do note that our method can be used with any other contrastive learning objective that allows the computation of a sample-wise losses.

SimSiam works as follows. Assume a given set of images $\mathcal{D}$, an image augmentation distribution $\mathcal{T}$, a minibatch of $M$ images $\mathbf{x} = \{x_i\}_{i=1}^{M}$ sampled uniformly from $\mathcal{D}$, and two sets of randomly sampled image augmentations $A$ and $B$ sampled from $\mathcal{T}$. SimSiam applies $A$ and $B$ to each image in $\mathbf{x}$ resulting in $\mathbf{x}^A$ and $\mathbf{x}^B$. Both augmented sets of views are subsequently projected into an embedding space with $\mathbf{z}^A = g_\theta(f_\theta(\mathbf{x}^A))$ and $\mathbf{h}^B = f_\theta(\mathbf{x}^B)$ where $f_\theta$ represents an encoder (or backbone) and $g_\theta$ a projector network. SimSiam then minimizes the following objective:

$$\mathcal{L}(\theta) = \frac{1}{2}\left(D(\mathbf{z}^A, \mathbf{h}^B) + D(\mathbf{z}^B, \mathbf{h}^A)\right) \tag{1}$$

where $D$ denotes the negative cosine similarity function. Intuitively, when optimizing $\theta$, the embeddings of the two augmented views are attracted to each other.

## 3.2 Hard View Selection

As previously described, Hard View Selection extends the random view generation by sampling adversarially harder views during pretraining. Instead of having two sets of augmentations $A$ and $B$, we now sample $N$ sets of augmentations, denoted as $\mathcal{A} = \{A_1, A_2, \ldots, A_N\}$. Each set $A_i$ is sampled from the image augmentation distribution $\mathcal{T}$, and applied to each image in $\mathbf{x}$, resulting in $N$ augmented sets of views $\mathbf{x}^{A_1}, \mathbf{x}^{A_2}, \ldots, \mathbf{x}^{A_N}$. Similarly, we obtain $N$ sets of embeddings $\mathbf{z}^{A_1}, \mathbf{z}^{A_2}, \ldots, \mathbf{z}^{A_N}$ and predictions $\mathbf{h}^{A_1}, \mathbf{h}^{A_2}, \ldots, \mathbf{h}^{A_N}$ through the encoder and projector networks. We then define a new objective function that seeks to find the pair $(x_i^{A_k}, x_i^{A_l})$ of a given image $x_i$ that yields the highest loss:

$$(x_i^{A_k}, x_i^{A_l}) = \underset{k,l;k\neq l}{\arg\max}\, \mathcal{L}(\theta)_{i,k,l} = \underset{k,l;k\neq l}{\arg\max}\, \frac{1}{2}\left(D(z_i^{A_k}, h_i^{A_l}) + D(z_i^{A_l}, h_i^{A_k})\right), \tag{2}$$

where $\mathcal{L}(\theta)_{i,k,l}$ simply denotes a sample-wise variant of Eq. 1. Overall, we first generate $N$ augmented views for each image $x_i$ in the minibatch. Then, we create all combinatorially possible $\binom{N}{2}$ pairs of augmented images for $x_i$. Then, we use Eq. 2 to compute the sample-wise loss for each pair which corresponds to invoking forward passes. We then select all pairs that yielded the highest loss to form the new *hard* minibatch of augmented sets $\mathbf{x}^{A_{k*}}$ and $\mathbf{x}^{B_{l*}}$, discard the other pairs and use the hard minibatch for optimization.

As shown in Algorithm 1, we repeat this process in each training iteration. This modification introduces a more challenging learning scenario in which the model is encouraged to learn more discriminative features by being exposed to harder views. While the additional forward passes come with a certain overhead (about a factor of $1.55\times$ for SimSiam), as we will demonstrate in our experiments, the focus on harder views leads to consistent improvements.

---

**Algorithm 1** Contrastive Learning with HVS

1: **Input:** Number of views $N \geq 2$, batch size $M$,
2: augmentation distribution $\mathcal{T}$, encoder $f_\theta$, projector $g_\theta$
3: **for** each $x_i$ in the sampled minibatch $\{x_i\}_{i=1}^{M}$ **do**
4:      Sample $N$ augmentations: $A = \{t_n \sim \mathcal{T}\}_{n=0}^{N}$
5:      Create augmented views: $\mathbf{x}_i^A = \{t_n(x_i)\}_{n=0}^{N}$
6:      Forward all views through $f_\theta$ and $g_\theta$ to get $z$ and $h$
7:      Create all $\binom{N}{2}$ possible pairs $\mathbf{x}_i^{A_k} \times \mathbf{x}_i^{A_l}$, $k \neq l$
8:      Select the *hard* pair $(x_i^{A_{k*}}, x_i^{A_{l*}})$ maximizing Eq. 2
9: Create the minibatch $(\mathbf{x}^{A_{k*}}, \mathbf{x}^{A_{l*}})$ consisting of hard pairs
10: Proceed with standard contrastive learning training to update $\theta$ with the hard batch
11: Repeat for all minibatches
12: **return** Optimized $\theta$

---

While we exemplified the integration of HVS with the SimSiam objective, integrating it into other contrastive methods is as straightforward. The only requirement of HVS is to be able to compute sample-wise losses (in order to select the views with the highest loss). In our experiments section and in addition to SimSiam, we study the application of HVS to the objectives of DINO and SimCLR (see also Appendix I.1 for a formal exemplary definition for the integration of HVS into SimCLR).

## 3.3 Implementation and Evaluation Protocols

**Implementation** We now describe the technical details of our approach. HVS can be used with any contrastive method that allows computing sample-wise losses, and the only two elements in

the pipeline we adapt are: 1) the data loader (needs to sample $N$ views for each image) and 2) the forward pass (invokes the *select* function to identify and return the hard views). The image transformation distribution $\mathcal{T}$ of the baselines is left unchanged. Note, for the view selection one could simply resort to RRC only and apply the rest of operations after the hard view selection (see Section 5.1 for a study on the influence of appearance on the selection). All experiments were conducted with $N = 4$ sampled views, yielding $\binom{N}{2} = 6$ pairs to compare, except for DINO that uses 10 views (2 global, 8 local heads) per default. For DINO, we apply HVS to both global and local heads but to remain tractable, we upper-bound the number of total pair comparisons to 128. SimCLR uses positive and negative samples. In accordance with the simplicity of HVS, we do not alter its objective, which naturally leads to selecting hard views that are adversarial to positive and "cooperative" to negative views.

**Evaluation Protocols**   We now describe the protocols used to evaluate the performance in our main results section. In self-supervised learning, it is common to assess pretraining performance with the linear evaluation protocol by training a linear classifier on top of frozen features or finetuning the features on downstream tasks. Our general procedure is to follow the baseline methods as closely as possible, including hyperparameters and code bases (if reported). It is common to use RRC and horizontal flips during training and report the test accuracy on central crops. Due to the sensitivity of hyperparameters, and as done by Caron et al. (2021), we also report the quality of features with a simple weighted nearest neighbor classifier (k-NN).

| Method | Arch. | 100 epochs | | 300 epochs | |
|---|---|---|---|---|---|
| | | Lin. | $k$-NN | Lin. | $k$-NN |
| DINO | ViT-S/16 | 73.52 | 68.80 | 75.48 | 72.62 |
| + HVS | ViT-S/16 | 74.67 | 70.72 | 76.56 | 73.64 |
| **Improvement** | | **+1.15** | **+1.92** | **+1.08** | **+1.02** |
| DINO | RN50 | 71.93 | 66.28 | 75.25 | 69.53 |
| + HVS | RN50 | 72.87 | 67.33 | 75.65 | 70.05 |
| **Improvement** | | **+0.94** | **+1.05** | **+0.40** | **+0.52** |
| SimSiam | RN50 | 68.20 | 57.47 | 70.35 | 61.40 |
| + HVS | RN50 | 68.98 | 58.97 | 70.90 | 62.97 |
| **Improvement** | | **+0.78** | **+1.50** | **+0.55** | **+1.57** |
| SimCLR | RN50 | 63.40 | 52.83 | 65.50 | 55.65 |
| + HVS | RN50 | 65.33 | 54.76 | 67.20 | 57.31 |
| **Improvement** | | **+1.93** | **+1.85** | **+1.70** | **+1.66** |
| iBOT | ViT-S/16 | 69.50 | 62.82 | 72.48 | 66.60 |
| + HVS | ViT-S/16 | 70.41 | 63.05 | 73.99 | 67.12 |
| **Improvement** | | **+0.91** | **+0.23** | **+1.51** | **+0.52** |

Table 1: Average top-1 linear and $k$-NN classification accuracy on the ImageNet validation set for 100 and 300-epoch pretrainings. For SimCLR we run 200 instead of 300 epochs.

## 4   MAIN RESULTS

Here, we discuss our main results on image classification, object detection, and segmentation tasks. All results are self-reproduced using the baseline code and available hyperparameters, and are averaged over 3 seeds (except for iBOT which is averaged over 2 seeds). All hyperparameters are reported in Appendix G.

### 4.1   EVALUATIONS ON IMAGENET

We report the top-1 validation accuracy on frozen features, as well as the k-NN classifier performance, in Table 1. For DINO, we additionally compare ResNet-50 (He et al., 2016) against the ViT-S/16 (Dosovitskiy et al., 2020) architecture. Our method compares favourably against all baselines with an increased performance of approximately $1\%$ on average, showing the benefit of sampling hard views. We highlight that our 300-epoch DINO+HVS model exhibits strong performance, closely rivaling the officially reported 800-epoch DINO performance of 77%, with a mere $\sim 0.4\%$ performance gap. This achievement remains favorable, even when factoring in the $2\times$ slowdown (DINO with 2 global, 8 local heads) induced by our method. Due to limited compute resources, we run the pretrainings for 100 and 300 epochs (200 epochs for SimCLR) and batch sizes of 512 (100 epoch) or 1024 (200 & 300 epoch trainings), respectively. This choice is in line with a strategy that favors the evaluation of a diverse and larger set of baselines over the evaluation of a less diverse and smaller set and underpins the broad applicability of HVS. We primarily ran our experiments with 8xNVIDIA GeForce RTX 2080 Ti nodes, with which the cheapest runs required $\sim 3.5$ days for pretraining and linear evaluation and the most expensive runs required $\sim 25$ days. Moreover, we emphasize that HVS is insensitive to the baseline hyperparameters and simply reusing these consistently resulted in improvements of the reported magnitudes across all experiments.

| Method | Arch. | CIFAR10 | | CIFAR100 | | Flowers102 | | iNat 21 | | Food101 | |
|--------|-------|------|------|------|------|------|------|------|------|------|------|
| | | Lin. | F.T. | Lin. | F.T. | Lin. | F.T. | Lin. | F.T. | Lin. | F.T. |
| SimSiam | RN50 | 82.60 | 95.50 | 54.20 | 77.20 | 34.27 | 56.40 | 32.50 | 60.30 | 65.70 | 83.90 |
| + HVS | RN50 | 84.40 | 96.10 | 57.10 | 78.20 | 38.37 | 58.90 | 33.90 | 60.90 | 67.10 | 84.70 |
| Improvement | | **+1.80** | **+0.60** | **+2.90** | **+1.00** | **+4.10** | **+2.50** | **+1.40** | **+0.60** | **+1.40** | **+0.80** |
| DINO | ViT-S/16 | 94.53 | 98.53 | 80.63 | 87.90 | 91.10 | 93.20 | 46.93 | 53.97 | 73.30 | 87.50 |
| + HVS | ViT-S/16 | 95.13 | 98.65 | 81.27 | 88.23 | 92.07 | 93.60 | 49.03 | 54.16 | 74.13 | 87.91 |
| Improvement | | **+0.60** | **+0.12** | **+0.63** | **+0.33** | **+0.97** | **+0.40** | **+2.10** | **+0.19** | **+0.83** | **+0.41** |

Table 2: Models trained on ImageNet with HVS compare favourably against models trained without it when transferred to other datasets through finetuning (F.T.) or the linear evaluation protocol (Lin.).

## 4.2 TRANSFER TO OTHER DATASETS AND TASKS

We now demonstrate the transferability of features learned with HVS. For all transfer experiments, we use our 100-epoch ImageNet pretrained SimSiam ResNet-50 and DINO ViT-S/16 models.

**Linear Evaluation and Finetuning** In Table 2, we apply both the linear evaluation (Lin.) and finetuning (F.T.) protocols to our models across a diverse set of datasets consisting of CIFAR10 Krizhevsky (2009), CIFAR100, Flowers102 Nilsback & Zisserman (2008), Food101 Bossard et al. (2014), and iNaturalist 2021 (iNaturalist 2021 competition dataset). Our results show that the improvements achieved by sampling hard views we observed so far also transfer to other datasets.

**Object Detection and Instance Segmentation** For object detection, we use the VOC07+12 (Everingham et al., 2010) dataset using Faster R-CNN (Ren et al., 2015). We study instance segmentation on the COCO (Lin et al., 2014) dataset using Mask R-CNN (He et al., 2022). Table 3, where we report the AP50 performance, shows that the features learned with HVS also mostly transfer favourably to these tasks and outperform the SimSiam baseline. However, in case of DINO, we observe a slight performance gap on the detection task, which is the only instance in the entire paper where HVS did not outperform the baseline. This result warrants further investigation, particularly considering that this experiment was conducted with a single seed (due to limited availability of compute resources). More details and performance results on this task are provided in Appendix F.1.

| Method | Arch. | VOC Det. | COCO Segm. |
|--------|-------|----------|------------|
| SimSiam | RN50 | 78.46 | 45.16 |
| + HVS | RN50 | 79.06 | 46.51 |
| **Impr.** | | **+0.60** | **+1.35** |
| DINO | RN50 | 80.86 | 50.35 |
| + HVS | RN50 | 80.51 | 50.39 |
| **Impr.** | | **-0.36** | **+0.04** |

Table 3: AP50 for object detection and instance segmentation with 100-ep. pretraining.

## 5 EMPIRICAL ANALYSIS OF HARD VIEW SELECTION

In this section, we discuss studies designed to shed light on the mechanisms underlying HVS. We address the following questions: 1) "Which can be observed that underlie the hard view selection?" 2) "Can a "manual" augmentation policy be inferred from these patterns?", and 3) "What are the effects on empowering the adversary?". For all experiments conducted here, we use our 100-epoch, ImageNet-pretrained SimSiam+HVS models with four sampled views.

### 5.1 Q1: WHICH PATTERNS UNDERLYING THE HARD VIEW SELECTION CAN BE OBSERVED?

When visually studying some of our examples and the views selected by HVS in Figures 2 and 6 (in the appendix), we notice that both geometric and appearance characteristics seem to be exploited. E.g., consider the brightness difference between the views of the first two rows in Fig. 2. Additionally, we observe a generally higher training loss (Fig. 7 in the appendix) indicative of an increased task difficulty.

**Logging Augmentation Data** To assess these observations quantitatively, we logged relevant hyperparameter data during SimSiam training with HVS and conducted multiple studies which we

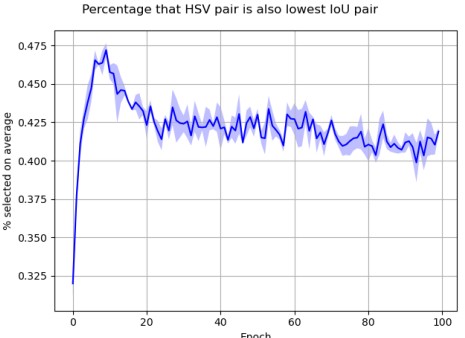 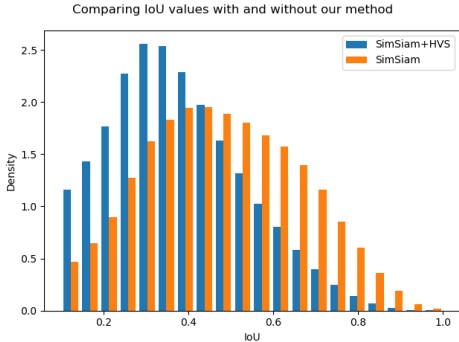

Figure 3: **Left:** In over 40% of the cases, the adversarially selected view pair is also the one with the lowest IoU throughout SimSiam+HVS pretraining. We attribute the spike in the early phase to the random initialization of the embedding. **Right:** When using HVS (blue), a shift to smaller IoU values is visible when comparing to standard pretraining (orange). Both results are based on 3 seeds.

will describe below. The data we logged include for each view the sampled parameters from the geometric and appearance data augmentation operations (such as the height/width and coordinates of the crops) as well as color distortion parameters (such as brightness and contrast, see Section D in the appendix for more details), as well as the loss and whether the view was selected or not. As evaluated metrics we chose the Intersection over Union (IoU), Relative Distance (image-wise normalized distance of the center points of view pairs), color distortion distance (the Euclidean distance between all four color distortion parameters), and the individual color distortion parameters.

**Importance of Augmentation Metrics**   Given 300k such samples, we then used fANOVA (Hutter et al., 2014) to determine how predictive these metrics are. This resulted in the metric with the highest predictive capacity on the loss to be the IoU, explaining 15% of the variance in performance, followed by brightness with 5% (for more details see Fig. 9 in the appendix). The importance of IoU in HVS is further underpinned by the following observation: the fraction of view pairs selected by HVS, which are also the pairs with the lowest IoU among all six pairs (N=4), is over 40% (random: ~16.7%) during training. Moreover, when using HVS, a shift to smaller IoU values can be observed when comparing against standard pretraining. We visualize both findings in Fig. 3.

**Taking a Closer Look at the Intersection over Union**   We also examined the IoU value over the course of training in Fig. 4 (left). Here, we notice the average IoU value of selected view pairs increases slightly with training, possibly as a reaction of the pretrained model's embedding representation to the hard view selection. Another pattern we observe is that the IoU value with HVS (Fig. 4 (left) in blue) is smaller and also less constant during training when compared against training without HVS (green). We believe this is due to the sample-wise and stateful nature of the adversarial selection as HVS chooses different IoU values from sample to sample and training state to training state. Lastly, we assessed the effect of the color augmentation on the pair selection. For this study, we sampled *one* set of color augmentations (as opposed to multiple, i.e. one each view) per iteration and applied it to all views. Only after identifying the hardest pair, we apply sampled data augmentations for each view individually. Consequently, the linear evaluation performance dropped by 0.3% on average. Interestingly, as we show in Fig. 4 (right), the fraction of selected pairs that are also the hardest pairs slightly increases in this case. One possible explanation for this is that it reflects the non-negligible role of color variation between views (as shown previously with the importance analysis), where HVS is given less leverage to increase hardness through a static appearance and instead, depends more on leveraging the IoU. Lastly, another key observation is that HVS often chooses view pairs that incorporate zooming in and out or an increased distance between the views (see last row of Fig. 2). In our view, the former relates well to the analogy of local-to-global view correspondence that was first introduced in SimCLR and further expanded on in DINO.

## 5.2   Q2: CAN A MANUAL AUGMENTATION POLICY BE INFERRED?

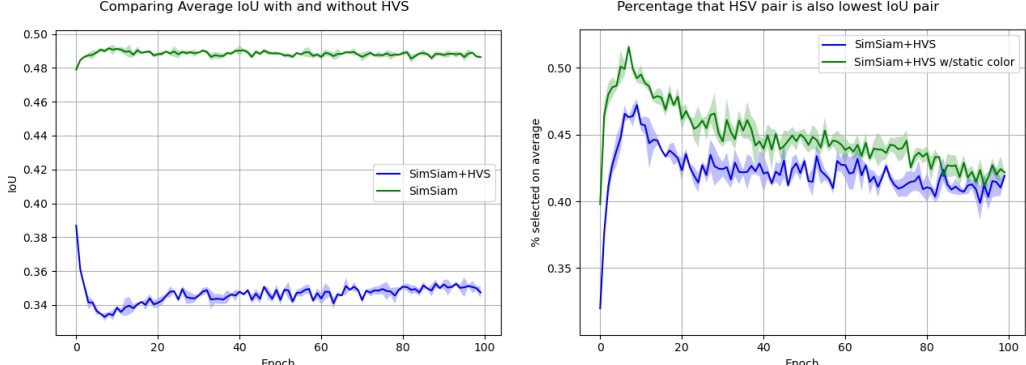

Figure 4: **Left:** The average IoU of view pairs selected by SimSiam+HVS (blue) compared against the default SimSiam training (green). **Right:** Using static color augmentation for all pairs before the selection increases the dependency on the IoU.

Since harder pretraining tasks seem beneficial, and based on the observations made in Q1, a natural question arises: is it possible to mimic the adversarial selection with a manually scripted augmentation policy? Such a policy would replace HVS and lower the computational cost by avoiding the additional forward passes. Since the IoU seems to play an important role, below, we study several possible ways for constructing a simple manual augmentation policy based on IoU.

**Deriving an Augmentation Policy** We implemented the following algorithm based on rejection sampling in the augmentation pipeline: we linearly approximate the IoU value curve from Fig. 4 (left; in blue) with a start value of 0.30 and end value 0.35 (ignoring the dip in the early phase) to receive an upper bound for the IoU in each training iteration. For each iteration, we then check if the pairs exceed the IoU upper bound value and if so, we reject the pair and re-sample a new pair (max. 512 times). This ensures that only view pairs are sampled that employ at least

| IoU Policy Type | SimSiam | DINO |
|---|---|---|
| Baseline (B) | 68.20 | 73.50 |
| B+range(0.3-0.35) | -0.80 | -1.47 |
| B+range(0.3-0.35)+alternating | +0.10 | -0.45 |
| B+range(0.4-0.6) | +0.55 | -0.40 |
| B+range(0.4-0.6)+alternating | +0.25 | -0.20 |
| B+range(0.1-0.8) | -33.95 | -1.50 |
| B+range(1.0-0.1) | +0.07 | - |

Table 4: Top-1 lin. eval. accuracies for the manual IoU policy (averaged over 2 seeds).

a certain task difficulty (by means of a small enough IoU). We varied different hyperparameters (e.g., IoU start/end ranges; inverse schedules; alternating between the IoU schedule and the standard augmentation every other iteration; as well as warmup, i.e. no IoU policy for the first 10 epochs). Training both SimSiam and DINO (ViT-S/16; applied to all global and local heads) models for 100 epochs yielded performance decreases or insignificant improvements, as can be seen in Table 4. Moreover, we see an increased risk of model collapse when setting start values too small which can be interpreted as a too hard task. These results indicate that developing a manual policy based on metrics in pixel space such as the IoU is non-trivial. Additionally, these results show that transferring such a policy from SimSiam to DINO does not work well, possibly due to additional variations in the augmentation pipeline such as multi-crop. In contrast, we believe that HVS is effective and transfers well since it 1) operates on a similarity level of latent embeddings that may be decorrelated from the pixel space and 2) has access to the current model state.

**Assessing the Difficulty of Predicting the Pair Selected by HVS** To further validate the previous result and to ascertain an upper limit on the performance for predicting the hardest pair, we conducted a second experiment. We fitted a gradient boosting classification tree (Friedman, 2001) to predict the selected view pair conditioned on all SimSiam hyperparameter log data from Q1 (see above) except for the flag that indicates whether a view was selected. As training and test data, we used the logs from two seeds (each 300k samples) and the logs from a third seed, respectively. We also tuned hyperparameters on train/valid splits and applied a 5-fold CV. However, the resulting average test performance in all scenarios never exceeded 40%, indicating that it is indeed challenging to predict the hardness of views based on parameter-level data. Consequently, we believe these results support the presumption that deriving a policy for controlling and increasing hardness based on geometric and appearance parameters is non-trivial. Furthermore, our analysis suggests that such

a policy must function on a per-sample basis and have access to the current model state which is the case when applying HVS.

### 5.3 Q3: What are the Effects of Empowering the Adversary?

It is well known that adversarial learning can suffer from algorithmic instability (Xing et al., 2021), e.g. by giving an adversary too much capacity. We now further explore the space of adversarial capacity for selecting hard views to gain a better understanding of its impact on contrastive learning.

**Optimizing an Adversarial Learner for the View Generation**  In this experiment, we briefly explore adversarially learning a network to output the transformation matrix for view generation. We optimize a Spatial Transformer Network (STN) (Jaderberg et al., 2015) jointly with the DINO objective and a ViT-tiny. Using the inverse of the DINO loss, we train the STN to output the geometric transformation matrices (each 6 parameters) for generating views given an input image that are used for standard CL training with DINO. We employ and test various transformation types (e.g., rotation, translation, etc.) that influence the capacity of the adversary. We discuss this experiment in greater detail in Appendix E and briefly summarize our findings here: applying an adversary in this way is very challenging due to the sensitivity of CL on the data augmentation and the dynamics of adversarial learning, often leading to either minor improvements or model collapses as we have observed when employing a too strong manual IoU policy in Q1.

**Increasing the Number of Views**  In our initial experiments, we explored variations in the number of sampled views $N$ with SimSiam and HVS. As can be seen in Fig. 8 in the appendix, while $N = 8$ views still outperform the baseline in terms of linear evaluation accuracy, it is slightly worse than using $N = 4$ views (-0.05% for 100 epochs and -0.14% for 200 epochs pretraining on linear evaluation). We interpret this result as the existence of a "sweet spot" in setting the number of views, where, in the limit, a higher number of views corresponds to approximating the adversarial learner described above. This would allow HVS to select progressively harder views that quickly results in very hard tasks and consequently in model collapse and performance deterioration. [1]

**Longer Training**  Despite the inherent overhead associated with additional forward passes of HVS, our results clearly show that using HVS is also beneficial for long trainings (see 300-epoch improvements in Table 1). To delve deeper into the behavior of longer trainings under our computational constraints, we conduct several training runs with progressively higher epochs on the CIFAR100 dataset, using the same initial learning rate and a cosine schedule. The comparative analysis in Fig. 5 shows, except for some overfitting, no evidence of diminishing returns as the improvement remains consistent throughout training nor signs of instability (e.g., collapse) due to hard tasks.

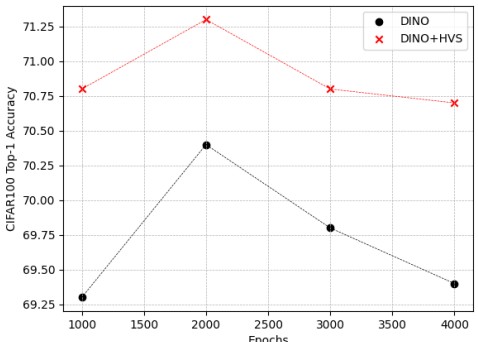

Figure 5: Comparing DINO with DINO+HVS on CIFAR100 for longer trainings.

## 6 Conclusion

In this study, we presented HVS, a new data augmentation and learning strategy for contrastive learning designed to challenge pretrained models with more demanding samples. This method, while straightforward in its design, has proven to be a powerful tool, pushing the boundaries of the traditional random view generation in contrastive learning. When combined with methods like DINO, SimSiam, and SimCLR, HVS consistently showcased improvements in linear evaluation and a diverse set of transfer tasks. As the landscape of self-supervised learning continues to evolve, methods like HVS underscore the potential for further advancements in the field. By continuously

---

[1]For completeness, we also experimented with a cooperative HVS (see Appendix F.2 for details).

challenging models with harder samples and refining augmentation techniques, we anticipate even greater strides in model performance and generalization in future research endeavors. While HVS incurs a computational overhead due to its additional forward passes, our preliminary experiments indicate no signs of diminishing returns with extended training durations. This suggests that HVS is likely to continue to outperform baselines regardless of the training length. Looking ahead, we believe there is ample room for refining and expanding HVS, such as further investigating the number of sampled views based on the current learning state, enhancing efficiency through checkpointing activations, bypassing forwarding of similar pairs, or augmentation distributions that allow controlling task difficulty.

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
