## APPENDIX A    EXAMPLES SAMPLED BY HVS

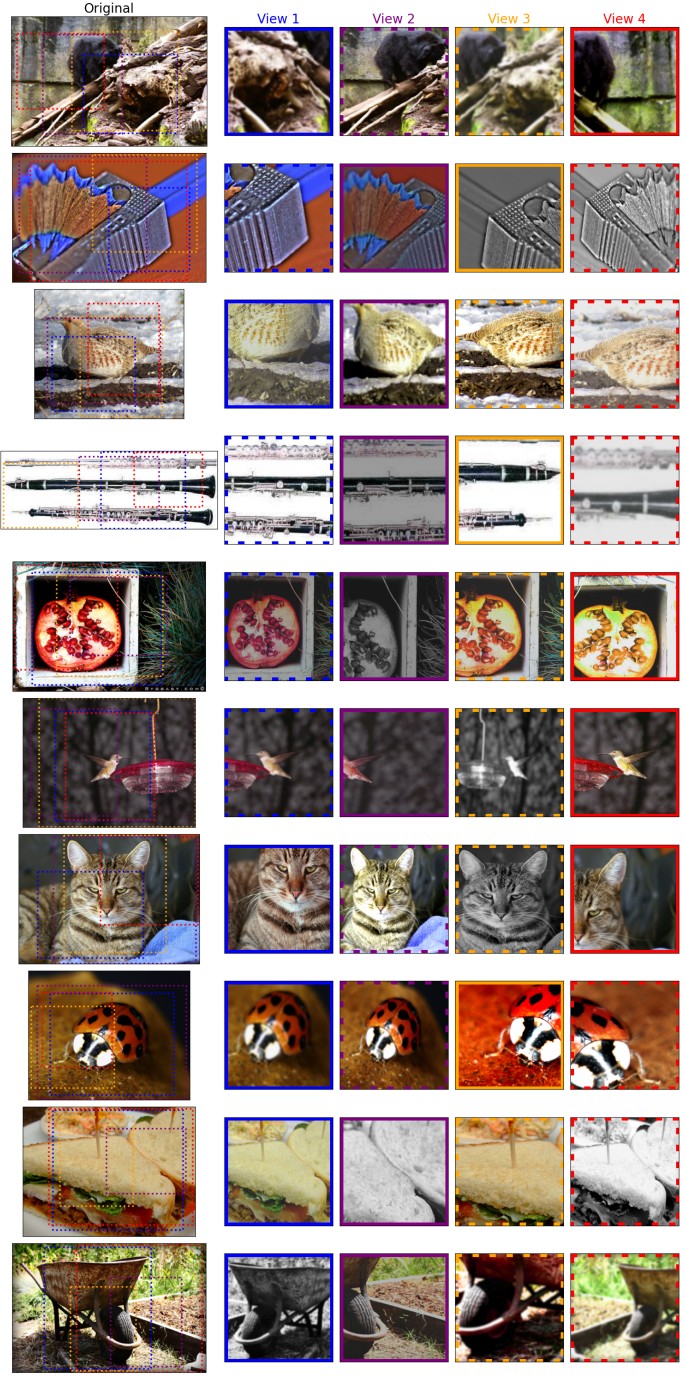

Figure 6: We depict row-wise ten example images from the ImageNet train set along with their sampled views with a finished, 100-epoch trained SimSiam ResNet50 model. Left: original image with the overlaid randomly sampled crops (colored dashed rectangles). Right: All views after applying resizing and appearance augmentations. The pair that is selected adversarially by HVS is highlighted in solid lines, eg. View 1 and View 4 in the first row.

## APPENDIX B    TRAINING LOSS

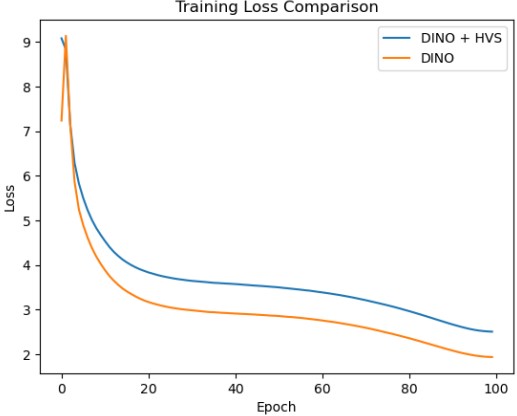

Figure 7: The training loss over 100 epochs. Comparing the DINO vanilla method with DINO + HVS. The spike and drop in the loss curve of DINO is caused by freezing the last layer in the first epoch which was proposed by the authors as a strategy to enhance downstream performance. For HVS we can only see a drop and no spike. We believe this is because HVS exposes the model to hard views from the beginning of training (i.e. the loss is immediately maximized).

## APPENDIX C    EFFECT OF MORE VIEWS ON LINEAR EVALUATION PERFORMANCE

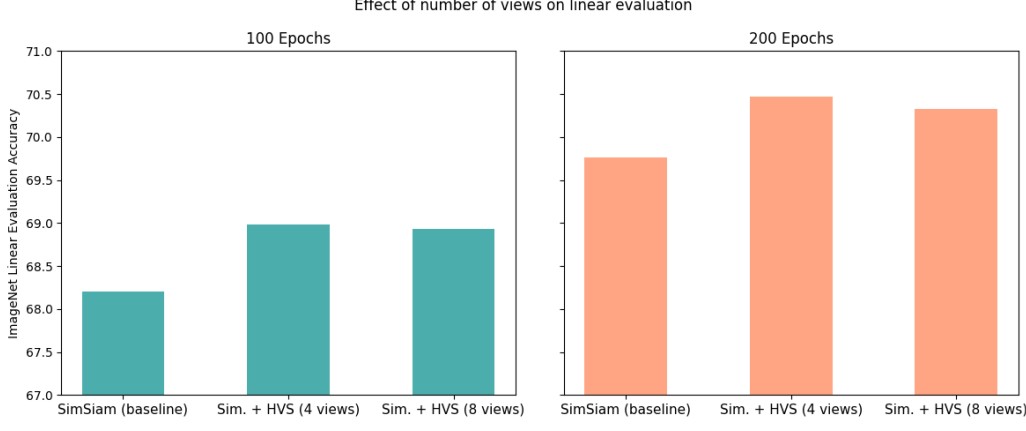

Figure 8: Setting the number of views too high can result in performance deterioration. This shows that diminishing returns exist, likely because the adversary becomes too strong, resulting in a too hard learning task.

## APPENDIX D    ASSESSING THE IMPORTANCE OF METRICS WITH *fANOVA*

To assess the importance of various metrics on the training loss, we apply fANOVA Hutter et al. (2014) on data that we logged during training with HVS. We used 300k samples that contain the following sampled parameters from the geometric and appearance data augmentation operations for each view: all random resized crop parameters (height and width of the original image, coordinates of crop corners and height and width of the crop), all Colorjitter (color distortion) strengths (brightness, contrast, saturation, hue), grayscale on/off, Gaussian blurring on/off, horizontal flip on/off,

loss, and if the crop was selected or not. The metrics we chose are Intersection over Union (IoU), Relative Distance (sample-wise normalized distance of the center points of crop pairs), color distortion distance (the Euclidean distance between all four color distortion operation parameters, i.e. brightness, contrast, saturation, hue), and the individual color distortion parameters of the Colorjitter operation. As can be seen in Fig. 9, the metric with the highest predictive capacity on the loss is the IoU with an importance of 15.2% followed by brightness with 5.1%. The relative distance has an importance of 3.3%, the colorjitter distance 2.3%, the contrast 1.6%, the saturation 1.4%, hue 0.6%, and all parameters jointly 1.7%.

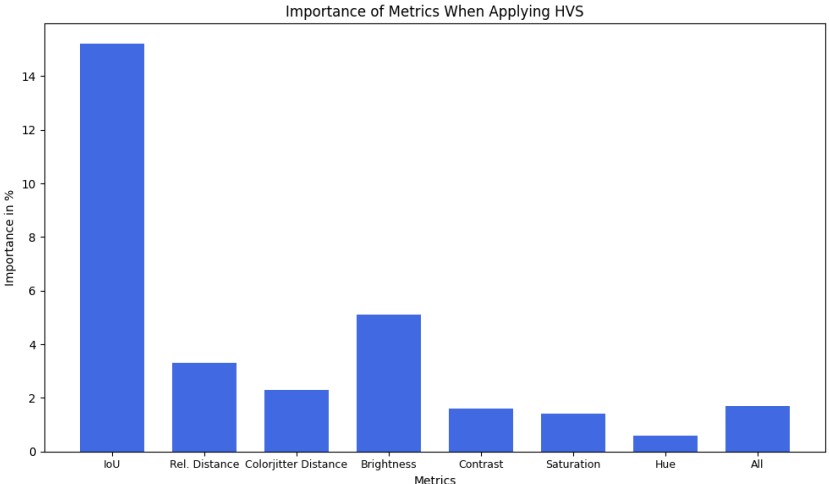

Figure 9: Application of fANOVA Hutter et al. (2014) on logged training data to determine metrics with high predictive capacity on the train loss.

## APPENDIX E    ADVERSARIAL LEARNER FOR THE VIEW GENERATION

We use Spatial Transformer Network (STN) (Jaderberg et al., 2015) to allow generating views by producing 6D transformation matrices (allowing translating, rotating, shearing, scaling, affine transformations and combinations thereof) in a differentiable way since most common augmentations are not off-the-shelf differentiable. As described, we train it alongside the actual pretrained network using the same (inverted) objective. For our experiments we use DINO with multi-crop, i.e. 2 global and 8 local heads. As STN we use a small CNN with a linear layer for outputting the 10*6D transformation matrices. In this scenario, we use a ViT-tiny/16 with a 300 epoch pretraining on CIFAR10 with a batch size of 256. All other hyperparameters are identical to the ones reported in the DINO paper.

Figure 10 visualizes the procedure. The STN takes the raw image input and generates a number of transformation matrices that are applied to transform the image input into views. These views are then passed to the DINO training pipeline. Both networks are trained jointly with the same loss function. DINO is trained with its original contrastive objective, where the STN is trained by inverting the gradient after the DINO during backpropagation.

As mentioned previously, the STN, without using auxiliary losses, starts zooming in and generating single-color views. To counteract this behavior, we experimented with different penalties on the transformation matrices produced by the STN. For instance, in order to limit the zooming pattern, we can use the determinants of the sub-matrices of the transformation matrix to penalize based on the area calculated and apply a regression loss (e.g. MSE). We refer to this type of penalty as *Theta Crops Penalty* (TCP). Additionally, we also restrict its parameters to stay within a sphere with different parameters for local and global crops. Next to determinant-based penalty losses, we also experimented with other penalty functions such as the weighted MSE between the identity and the current transformation matrix or penalties based on histograms of the input image and generated views after applying the transformation. To avoid strong uni-dimensional scaling behavior, we

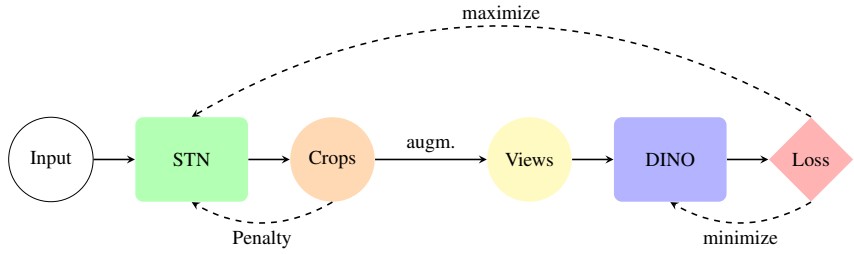

Figure 10: Illustration of adversarial learning with a Spatial Transformer Network (STN) jointly with contrastive learning (here: DINO).

also implemented restricting scaling in a symmetric way (i.e. applied to both x and y dimensions) and refer to this as *scale-sym.*. We report our best results in Table 5 which are all TCP-based. As can be seen, no setting is able to outperform the baseline. Our best score was achieved with translation-scale-symmetric which is very similar to random cropping. When removing the symmetries in scaling, the performance drops further. Removing a constraint adds one transformation parameter and therefore one dimension. This can be seen as giving more capacity to the adversarial learner which in turn can make the task significantly harder. Similarly, when adding rotation, the performance drops further and in part drastically. This is on the one hand due to the penalties not being fully able to restrict the output of the STN. On the other, the task of extracting useful information from two differently rotated crops is even harder, and learning spatial invariance becomes to too challenging. All in all, we experienced two modes: either the STN is too restricted, leading to *static output* (i.e. independent of image content, the STN would produce constant transformation matrices) or the STN has too much freedom, resulting in extremely difficult tasks. See Fig. 11 for an example on the former behavior.

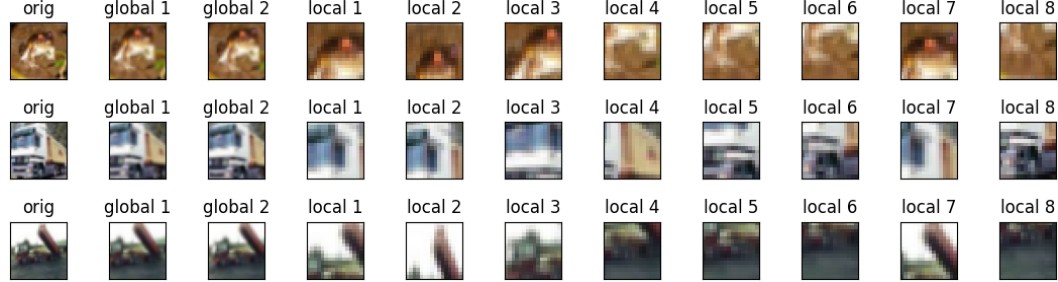

Figure 11: Example for static output behavior of the STN.

| Mode | Penalty | Lin. | F.T. |
|---|---|---|---|
| baseline | - | 86.1 | 92.7 |
| translation-scale-sym. | TCP | 83.7 | 90.3 |
| translation-scale | TCP | 82.8 | 89.7 |
| rotation-translation | TCP | 56.7 | - |
| rotation-translation-scale | TCP | 31.7 | - |
| rotation-translation-scale-sym. | TCP | 77.6 | - |
| affine | TCP | 78.3 | 83.5 |

Table 5: **Linear evaluation and finetuning classification performance on CIFAR10**. Top-1 accuracy on the validation set of CIFAR10 for our best results reported with different STN transformation modes.

# APPENDIX F  ADDITIONAL RESULTS

## F.1  OBJECT DETECTION AND INSTANCE SEGMENTATION

For our object detection and instance segmentation analysis experiment, we report additional results in Table F.1 based on the AP metric. We use the C4 backbone variant (Wu et al., 2019) and finetuning with the 1x schedule.

| Method | Arch. | VOC07+12 Object Det. | | | COCO Instance Segm. | | |
|---|---|---|---|---|---|---|---|
| | | $AP_{all}$ | $AP_{50}$ | $AP_{75}$ | $AP_{mask}$ | $AP_{mask50}$ | $AP_{mask75}$ |
| SimSiam | RN50 | 52.05 | 78.46 | 56.88 | 27.16 | 45.16 | 28.48 |
| + HVS | RN50 | 52.91 | 79.06 | 57.55 | 27.93 | 46.51 | 29.36 |
| **Improvement** | | **+0.86** | **+0.60** | **+0.67** | **+0.77** | **+1.35** | **+0.88** |
| DINO | RN50 | 53.66 | 80.86 | 58.60 | 30.25 | 50.35 | 31.78 |
| + HVS | RN50 | 52.86 | 80.51 | 58.21 | 30.02 | 50.39 | 31.52 |
| **Improvement** | | **+0.80** | **-0.36** | **-0.39** | **-0.23** | **+0.04** | **-0.27** |

Table 6:

## F.2  EASY VIEW SELECTION

To investigate the effect of a *cooperative*, i.e. easy pair selection, we conducted a small experiment. Instead of selecting the pair yielding the worst loss, we inverted the objective and selected the pair with the best loss. As expected, this led to model collapses with a linear eval. performance of 0.1%. This result is in line with previous findings that highlight the importance of strong augmentations in CL.

# APPENDIX G  HYPERPARAMETERS

## G.1  EVALUATIONS ON IMAGENET

### G.1.1  DINO

For DINO, we report the ViT pretraining hyperparameters in Table 7. For ResNet-50, we use the same hyperparameters. Note, for HVS we limit the total number of comparisons to 128 across all heads. Linear evaluation is executed for 100 epochs and we use a learning rate of 0.001, SGD optimizer (AdamW (Loshchilov & Hutter, 2019) during pretraining), a batch size of 1024, a momentum of 0.9, and no weight decay.

### G.1.2  SIMSIAM

In Table 8, we report the ResNet-50 pretraining hyperparameters. Linear evaluation is executed for 90 epochs (as reported by the SimSiam authors) and we use a learning rate of 0.1, LARS optimizer (You et al., 2017), a batch size of 4096, and no weight decay.

### G.1.3  SIMCLR

We report the ResNet-50 pretraining hyperparameters for SimCLR in Table 9. Linear evaluation is executed for 90 epochs with a learning rate 0.1, SGD optimizer, batch size of 4096 and no weight decay.

| Hyperparameter | Value | Hyperparameter | Value |
|---|---|---|---|
| architecture | vit_small | epochs: | 100 |
| img_size | 224 | warmup_epochs: | 10 |
| patch_size | 16 | freeze_last_layer: | 1 |
| out_dim | 65536 | lr: | 0.0005 |
| norm_last_layer | true | min_lr: | 1.0e-06 |
| momentum_teacher | 0.996 | optimizer: | AdamW |
| use_bn_in_head | false | weight_decay: | 0.04 |
| teacher_temp | 0.04 | weight_decay_end: | 0.4 |
| warmup_teacher_temp | 0.04 | global_crops_scale: | 0.4, 1.0 |
| warmup_teacher_temp_epochs | 0 | global_crops_size: | 224 |
| fp16 | true | local_crops_number: | 8 |
| batch_size | 512 | local_crops_scale | 0.05, 0.4 |
| clip_grad | 3.0 | local_crops_size: | 96 |
| drop_path_rate | 0.1 | | |

Table 7: Pretraining ImageNet hyperparameters for the runs with DINO. For 300 epochs, we use a batch size of 1024.

| Hyperparameter | Value |
|---|---|
| architecture | resnet50 |
| batch_size | 512 |
| blur_prob | 0.5 |
| crops_scale | 0.2, 1.0 |
| crop_size | 224 |
| feature_dimension | 2048 |
| epochs | 100 |
| fix_pred_lr | true |
| lr | 0.05 |
| momentum | 0.9 |
| predictor_dimension | 512 |
| weight_decay | 0.0001 |
| optimizer | SGD |

Table 8: Pretraining ImageNet hyperparameters for the runs with SimSiam. For 300 epochs, we use a batch size of 1024.

| Hyperparameter | Value |
|---|---|
| architecture | resnet50 |
| proj_hidden_dim | 2048 |
| out_dim | 128 |
| use_bn_in_head | true |
| batch_size | 4096 |
| optim | LARS |
| lr | 0.3 |
| sqrt_lr | false |
| momentum | 0.9 |
| weight_decay | 1e-4 |
| epochs | 100 |
| warmup_epochs | 10 |
| zero_init_residual | true |

Table 9:

### G.2 TRANSFER TO OTHER DATASETS AND TASKS

For linear evaluation on the transfer datasets, we simply used the same hyperparameters for linear evaluation on ImageNet (DINO and SimSiam respectively). For finetuning DINO ViT-S/16, we used the hyperparameters reported in Table 10 and for SimSiam ResNet-50 we used the hyperparameters in Table 11

| Hyperparameter | CIFAR10 | CIFAR100 | Flowers102 | iNat 21 | Food101 |
|---|---|---|---|---|---|
| lr | 7.5e-6 | 7.5e-6 | 5e-5 | 5e-5 | 5e-5 |
| weight_decay | 0.05 | 0.05 | 0.05 | 0.05 | 0.05 |
| optimizer | AdamW | AdamW | AdamW | AdamW | AdamW |
| epochs | 300 | 300 | 300 | 100 | 100 |
| batch_size | 512 | 512 | 512 | 512 | 512 |

Table 10: Finetuning hyperparameters for DINO ViT-S/16.

| Hyperparameter | CIFAR10 | CIFAR100 | Flowers102 | iNat 21 | Food101 |
|---|---|---|---|---|---|
| lr | 7.5e-6 | 5e-6 | 5e-4 | 7e-5 | 5e-6 |
| weight_decay | 0.05 | 0.05 | 0.05 | 0.05 | 0.05 |
| optimizer | AdamW | AdamW | AdamW | AdamW | AdamW |
| epochs | 300 | 300 | 300 | 100 | 100 |
| batch_size | 512 | 512 | 512 | 512 | 512 |

Table 11: Finetuning hyperparameters for SimSiam and ResNet-50.

### G.3 OBJECT DETECTION AND INSTANCE SEGMENTATION

We have used the Detectron2 library (Wu et al., 2019) for object detection and instance segmentation. We followed the public codebase from MoCo (He et al., 2020) (like SimSiam) for all entries. Due to limited compute resources we changed the batch size. All parameters that differ from MoCo are reported in the Table 12. The pretrained models are finetuned end-to-end on the target datasets. All methods are based on 100-epoch pre-training on ImageNet.

| | VOC07+12 Object Det. | Coce Inst. Segm. |
|---|---|---|
| Hyperparameter | Value | Value |
| batch_size | 8 | 8 |
| lr | 0.01 | 0.01 |
| steps | 48000 | 90000 |

Table 12: Hyperparameters for VOC object detection and COCO instance segmentation.

## APPENDIX H    COMPUTATIONAL OVERHEAD OF HVS

The computational overhead factors compared to baseline are as follows: SimCLR (x1.69), SimSiam (x1.55) and DINO (x2.15). For DINO's 2 global and 8 local views (default), applying HVS with nviews=2 sampled for each original view results in 4 global and 16 local views. Since considering all combinations would yield over 77k unique comparisons (4 over 2 times 16 over 8), to remain tractable, we limit the number of total comparisons to 128.

|  | SimCLR | SimSiam | DINO | iBOT |
|---|---|---|---|---|
| vs Vanilla | x1.69 | x1.55 | x2.15 | x1.50 |

Table 13: Time Overhead
Hardware/Software used: NVIDIA RTX 3080, AMD R7 5800X, 32GB RAM, Ubuntu 22.04,
PyTorch 2.0.1, CUDA 11.8

While technically there can be a memory overhead with HVS, with the number of sampled views chosen in this paper, the backward pass of the methods that compute gradients only for the selected view pair still consumes more memory than the selection part of HVS (even for 8 sampled views in SimSiam). Note, that selection and the backward computation are never executed at the same time but sequentially.

We emphasize that the time overhead factors were measured without any optimization of HVS' efficiency and, in our view, there are multiple ways to improve it. In our work, we opted for the easiest implementation possible to showcase that selecting harder views dependent on the model learning state can help boost performance in contrastive learning. Going forward with more compute-efficient HVS solutions, one could think of:

- using *smaller resolution views for the view selection, as done in the multi-crop (Caron et al., 2020) method used in DINO or
- using embeddings of views from "earlier" layers in the networks or
- using 4/8 bit low-precision for the view selection or
- using one GPU just for creating embeddings and selecting the hardest views while the remaining GPUs are used for learning or
- switching between HVS and the standard pipeline in alternating fashion or
- bypassing forwarding of similar pairs and more

*we tested briefly for DINO ViT-S/16 and SimSiam ResNet-50 the effect of halving the image resolutions:

- DINO ViT-S/16: **24% speed improvement** (0.51 versus 0.39s per batch) with global/local resolutions of $112^2$ (instead of $224^2$) and $48^2$ (instead of $96^2$)
- Simsiam RN50: **x% speed improvement** (x versus x) with resolution $112^2$ (instead of $224^2$)

Our central arguments can be summarized as follows:

1. There are little to no diminishing returns when training longer with HVS (seen for 300 epoch trainings on ImageNet and for 4000 epochs on CIFAR100 in Fig. 5).
2. When normalizing Table 1 with respect to training time, HVS still yields slightly better performances, even **without** any of the many possible efficiency improvements.

Given these arguments, we see our current investigation as an early-stage exploration that highlights the novelty and efficacy of selecting more challenging views based on the model's learning state for improving contrastive and non-contrastive learning performance. Our work may lay the groundwork for future explorations that could devise more efficient sampling methods to generate hard views and which in turn could benefit various SSL approaches.

## APPENDIX I   HARD VIEW SELECTION OBJECTIVES

### I.1   SIMCLR

In this section, we are going to introduce the application of HVS with the SimCLR objective. Assume a given set of images $\mathcal{D}$, an image augmentation distribution $\mathcal{T}$, a minibatch of $M$ images $\mathbf{x} = \{x_i\}_{i=1}^{M}$ sampled uniformly from $\mathcal{D}$, and two sets of randomly sampled image augmentations

$A = \{t_i \sim \mathcal{T}\}_{i=1}^M$ and $B$ sampled from $\mathcal{T}$. We apply $A$ and $B$ to each image in $\mathbf{x}$ resulting in $\mathbf{x}^A$ and $\mathbf{x}^B$. Both augmented sets of views are subsequently projected into an embedding space with $\mathbf{z}^A = g_\theta(f_\theta(\mathbf{x}^A))$ and $\mathbf{z}^B = g_\theta(f_\theta(\mathbf{x}^B))$ where $f_\theta$ represents an encoder (or backbone) and $g_\theta$ a projector network. Contrastive learning algorithms then minimize the following objective function:

$$\mathcal{L}(\mathcal{T}, \mathbf{x}; \theta) = -\log \frac{\exp(\mathrm{sim}(\mathbf{z}_i^A, \mathbf{z}_i^B)/\tau)}{\sum_{i \neq j} \exp(\mathrm{sim}(\mathbf{z}_i^A, \mathbf{z}_j^B)/\tau)} \tag{3}$$

where $\tau$ denotes a temperature parameter and *sim* a similarity function that is often chosen as cosine similarity. Intuitively, when optimizing $\theta$, embeddings of two augmented views of the same image are attracted to each other while embeddings of different images are pushed further away from each other.

To further enhance the training process, we introduce a modification to the loss function where instead of having two sets of augmentations $A$ and $B$, we now have "N" sets of augmentations, denoted as $\mathcal{A} = \{A_1, A_2, \ldots, A_N\}$. Each set $A_i$ is sampled from the image augmentation distribution $\mathcal{T}$, and applied to each image in $\mathbf{x}$, resulting in "N" augmented sets of views $\mathbf{x}^{A_1}, \mathbf{x}^{A_2}, \ldots, \mathbf{x}^{A_N}$.

Similarly, we obtain $N$ sets of embeddings $\mathbf{z}^{A_1}, \mathbf{z}^{A_2}, \ldots, \mathbf{z}^{A_N}$ through the encoder and projector networks defined as:

$$\mathbf{z}^{A_i} = g_\theta(f_\theta(\mathbf{x}^{A_i})), \quad i = 1, 2, \ldots, N$$

We then define a new objective function that seeks to find the pair of augmented images that yield the highest loss. The modified loss function is defined as:

$$\mathcal{L}_{\max}(\mathcal{T}, \mathbf{x}; \theta) = \max_{k,l:k \neq l} \mathcal{L}(\mathcal{T}, \mathbf{x}; \theta)_{kl}$$

where

$$\mathcal{L}(\mathcal{T}, \mathbf{x}; \theta)_{kl} = -\log \frac{\exp(\mathrm{sim}(\mathbf{z}_k^{A_k}, \mathbf{z}_k^{A_l})/\tau)}{\sum_{i \neq j} \exp(\mathrm{sim}(\mathbf{z}_i^{A_k}, \mathbf{z}_j^{A_l})/\tau)}$$

and $k, l \in \{1, 2, \ldots, N\}$ and $i, j \in \{1, 2, \ldots, M\}$.

For each iteration, we evaluate all possible view pairs and contrast each view against every other example in the mini-batch. Intuitively, the pair that yields the highest loss is selected, which is the pair that at the same time minimizes the numerator and maximizes the denominator in the above equation. In other words, the hardest pair is the one, that has the lowest similarity with another augmented view of itself and the lowest dissimilarity with all other examples.

## APPENDIX J    ATTENTION MAPS

In this section, we visualize attention maps as a way to qualitatively study the potential effects and differences of the HVS learning method and their resulting features. The provided attention maps are from a DINO ViT-S/16 100 model, and we contrast the attention for each HVS and the baseline. All input images are from the ImageNet-1k validation split. The color code used depicts strong attention in yellow and weak attention in green and attentions from the HVS model are shown in the top row and attentions from the baseline in the bottom row, respectively. To summrize this study, we do not see apparent strong differences between HVS and the baseline. What we occasionally notice is that the HVS models seem to capture the shape of some subtle/indistinct objects better (e.g. the creek/river in the valley in Fig. 13, the lizard in Fig. 14, or the speaker in Fig. 15 or focus slightly more on the context and background (Fig. 17, 18, and 19).

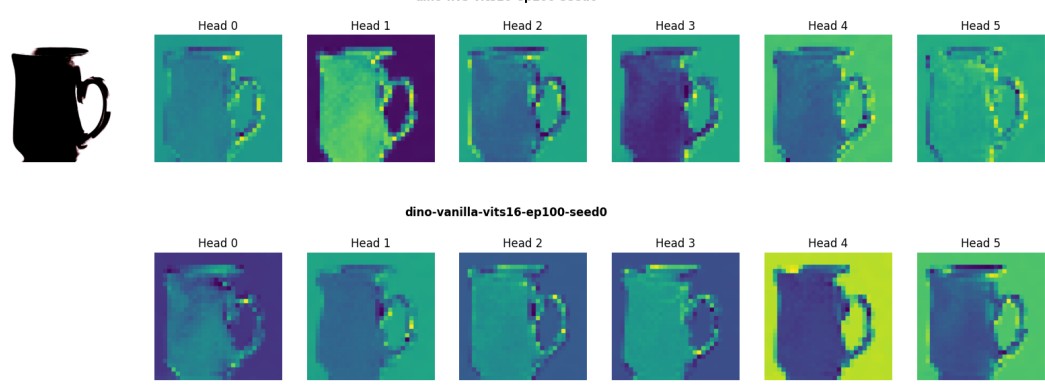

Figure 12:

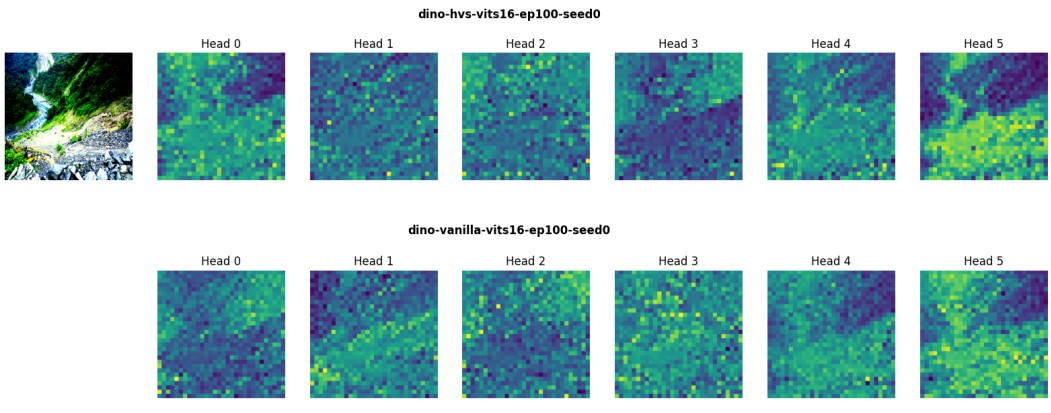

Figure 13:

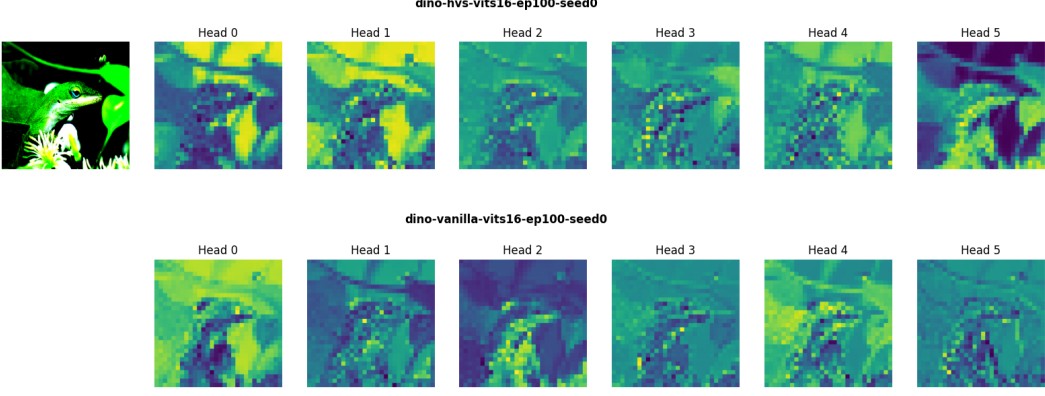

Figure 14:

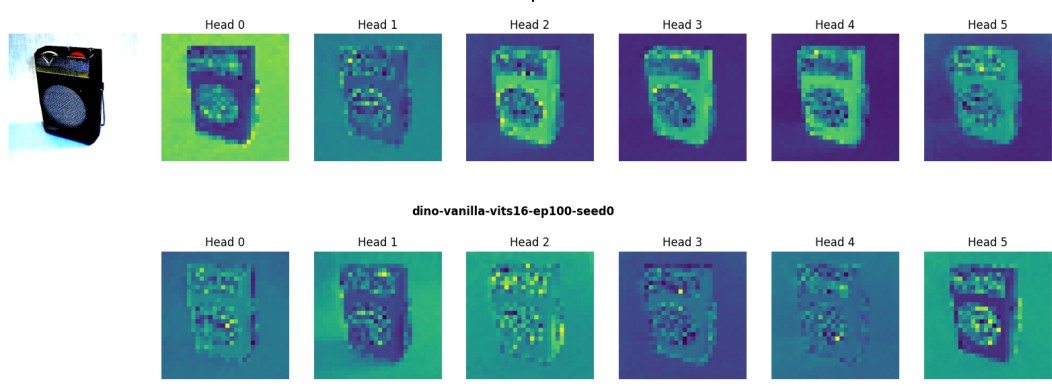

Figure 15:

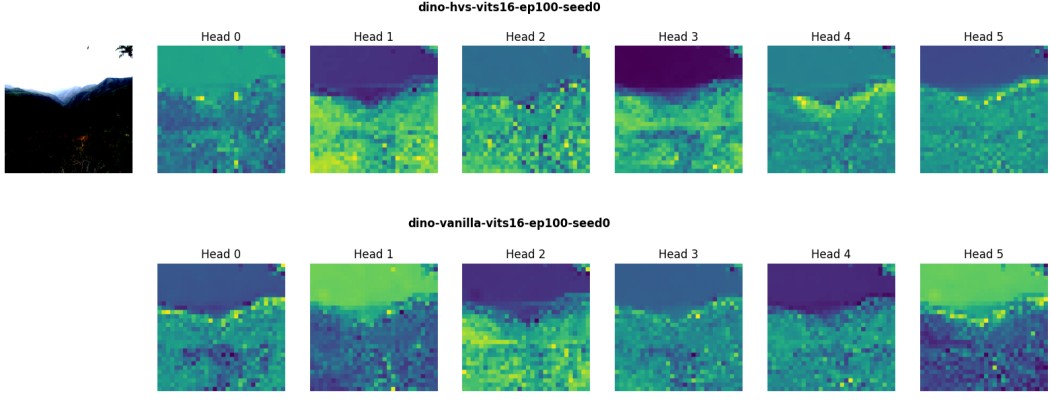

Figure 16:

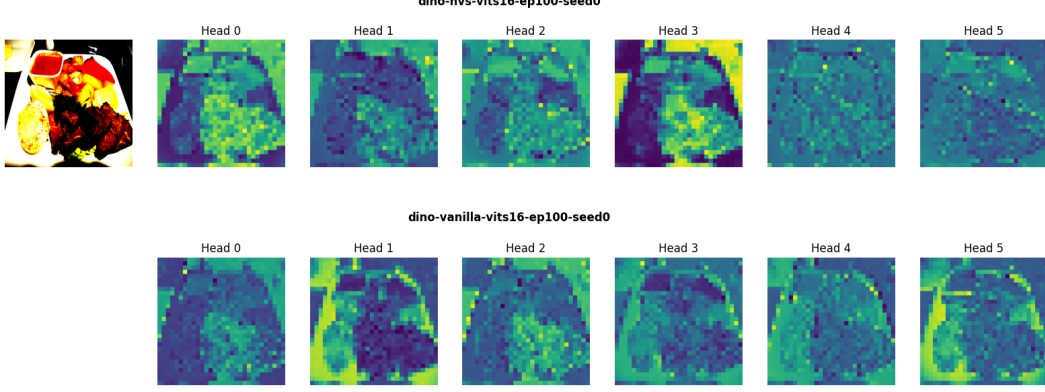

Figure 17:

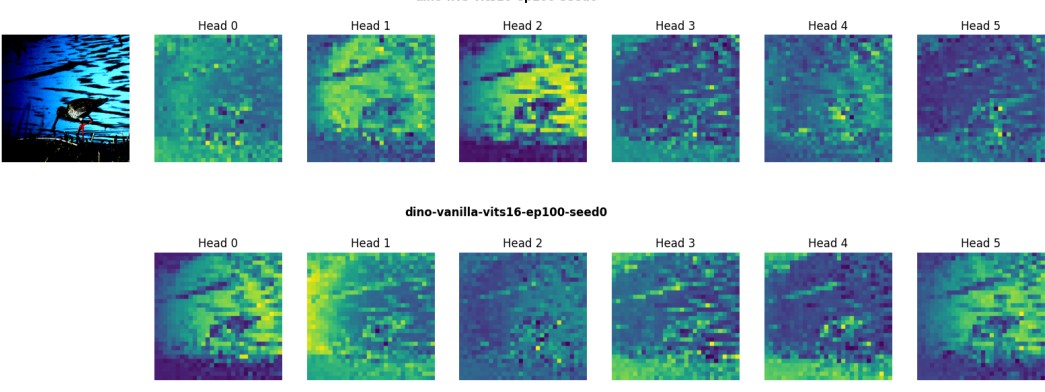

Figure 18:

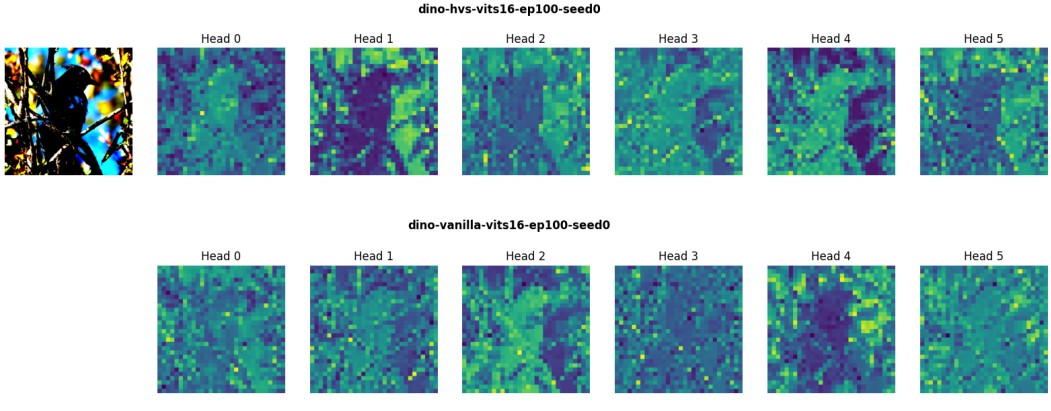

Figure 19:

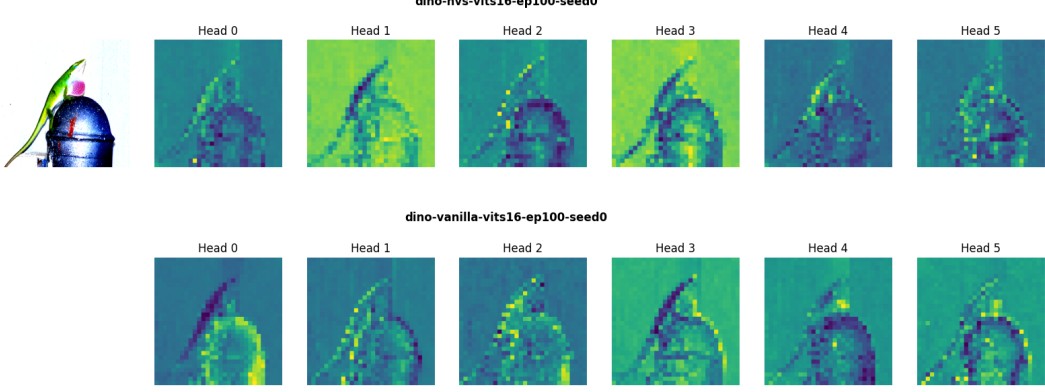

Figure 20: