# OpenReview forum: "Hard View Selection for Contrastive Learning"
_ICLR.cc/2024/Conference — Submitted to ICLR 2024_

### Official Review · Reviewer_zn9X · 2023-10-31

**Soundness:** 3 good
**Presentation:** 3 good
**Contribution:** 2 fair
**Rating:** 8
**Confidence:** 3

**Summary:**

The paper introduces Hard View Selection (HVS), aiming to improve the effectiveness of pretraining in contrastive learning scenarios. By selecting the "hardest" views during training, HVS pushes the model to learn more robust features. The authors claim compatibility with several popular contrastive learning methods, like SimSiam, DINO, and SimCLR, and demonstrate its effectiveness using ImageNet for pretraining.

**Strengths:**

- Novel Approach: The paper introduces a novel concept of "Hard View Selection (HVS)," which aims to improve the efficacy of pretraining in contrastive learning settings.
- Compatibility: One of the significant advantages of HVS is its compatibility with a variety of existing contrastive learning methods like SimSiam, DINO, and SimCLR. This makes it easily adaptable in various existing pipelines.
- Simplicity: The method is described as being simple to integrate, requiring only the ability to compute sample-wise losses. This lowers the bar for adoption and experimentation.
- Focus on Challenging Samples: By focusing on the "hardest" views during training, the paper takes an interesting approach to make the model focus on challenging aspects of the data, potentially leading to more robust feature learning.

**Weaknesses:**

- Computational Cost: The method involves selecting the "hardest" views by running forward passes for multiple view pairs, which could increase the computational cost of training, especially for large-scale datasets or complex models.
- Lack of Theoretical Analysis: The paper seems to focus on empirical validation but doesn't provide a theoretical foundation for why the "hard view selection" approach should work, which could limit its scientific rigor.
- Unclear Impact on Convergence: Introducing harder samples could potentially slow down the convergence of the training process

**Questions:**

Could you elaborate on the computational overhead introduced by the Hard View Selection (HVS) method, especially when dealing with large-scale datasets or complex models?

Could you provide more details on the complexities involved in computing sample-wise losses, particularly in the context of different neural network architectures?

Does the introduction of "harder" samples through HVS have any noticeable impact on the convergence speed or stability of the training process?

---

> ### Author Response · Authors · 2023-11-12
>
> **Lack of Theoretical Analysis: The paper seems to focus on empirical validation but doesn't provide a theoretical foundation for why the "hard view selection" approach should work, which could limit its scientific rigor.**
>
> Thank you for your feedback. We also thought about ways to incorporate a theoretical foundation and validation to our manuscript. Given the nature of our study however, we found it challenging to find an easy enough framework where math/theory could be applied, especially given the dynamics and assumptions of the various baselines HVS is applied to in our manuscript. Note also that, to our best knowledge, little to no theory exists for the considered baselines in this paper. We appreciate your point on enhancing scientific rigor and would be grateful for any suggestions or insights you might have on how to integrate theoretical analysis more effectively into our work.
>
> **Unclear Impact on Convergence: Introducing harder samples could potentially slow down the convergence of the training process**
>
> Across all our experiments, we fortunately observed no slow-down in convergence during training. More broadly speaking, the training loss pattern remained identical to the baseline training. The only difference to the baseline trainings we were able to observe is that the training loss is slightly higher (see Fig. 7 in the supplementary material for an example of the training loss with/without HVS). This might encourage better generalization.
>
> **Could you elaborate on the computational overhead introduced by the Hard View Selection (HVS) method, especially when dealing with large-scale datasets or complex models?**
>
> Thank you for communicating this concern. We have addressed this point in the common comment to all reviewers, and we hope that the additional information provided adequately addressed your inquiry. If you have any further questions regarding the computational overhead, please feel free to let us know.
>
> **Could you provide more details on the complexities involved in computing sample-wise losses, particularly in the context of different neural network architectures?**
>
> Since most contrastive/non-contrastive learning approaches preserve the sample-wise losses within a batch in their objective function, applying HVS works out of the box. In the rare cases where approaches (or architectures) collapse the batch (or sample) dimension, HVS cannot be applied out of the box. One example where this is the case is Barlow Twins [1]. Here, the authors choose an objective function where the cross-correlation matrix between embedding outputs is computed to minimize the distance to the identity cross-correlation matrix. During the computation of the cross-correlation, the batch-dimension is collapsed by matrix multiplication that results in loss of any sample-wise information. To still apply HVS here, one potential approach is to explore the construction of sub-matrices from the full cross-correlation matrix. These sub-matrices could be designed to correspond to individual samples, allowing for the recovery of sample-specific information. It's worth noting that, as of now, we haven't conducted tests to validate the effectiveness of this approach.
>
> [1] https://arxiv.org/abs/2103.03230
>
> **Does the introduction of "harder" samples through HVS have any noticeable impact on the convergence speed or stability of the training process?**
>
> When designing the HVS approach, we also thought of the potential risk of negative impacts with regard to convergence speed. However, empirically, we realized that HVS works extremely stably and has not impacted convergence speed across any of our experiments. Moreover, and as mentioned in the paper, it is also extremely robust to the baseline hyperparameters and does not require adaptation of these.
>
> We express our gratitude once more for the insightful feedback provided.

---

### Official Review · Reviewer_PWAF · 2023-11-01

**Soundness:** 2 fair
**Presentation:** 3 good
**Contribution:** 2 fair
**Rating:** 6
**Confidence:** 4

**Summary:**

The paper proposes an approach to improve self-supervised contrastive learning methods (computer vision) that aim to learn features invariant to different views (i.e. augmentation) of an input image. The paper introduces a simple (and easy to implement) algorithm “Hard View Selection (HSV)” for selecting the view pair used in SSL contrastive training. The goal of HSV algorithm is to select relatively hard views that improve the model's performance. HSV comprises of making forward calls with a set of randomly selected views, then making forward calls to pick out the view pair with highest loss for backpropagation step. The paper hypothesizes that selecting the view pair that are dependent on the current state of the model parameters yields the most benefits rather than using a fixed/random/learned adversary policy. The benefit of the proposed method is to empirically improve the SSL feature representation so that it could be show improvement for other downstream tasks (classification, detection, etc.). The paper shows that HSV is compatible with other contrastive methods like DINO, SimSiam and SimCLR.

**Strengths:**

The paper explores a strategy (HSV) for selecting the views (image augmentation pairs in contrastive learning) that leads to performance improvement on downstream tasks. HSV is a simple and easy to implement view-selection strategy that achieves improvement on ImageNet linear classification accuracy (order of 0.55% - 1.93%). The paper provides detailed ablation experiments to find an alternative view-sampling strategy (both learning-based eg. STN and statistics-based approach). Ablation experiments regarding the initial number of view pairs that determine the hardness of the selected view pair is also provided.(higher initial pool of view pairs would lead to more adversarial view selection for CL training). The paper is well-written and easy to follow. The paper provides code for reproducibility.

**Weaknesses:**

It would be helpful for the reader to get a better understanding of the following questions/suggestions:

1. A detailed analysis of the HVS overhead in terms of running time/ wall clock/ FLOPs/ throughput along with memory requirement for each the baselines DINO (2x overhead), SimSiam(1.55x overhead), SimCLR.  Preferably in a format of a table.

2. Extending on the above point, it seems since the overhead is added due to HSV, a more fair choice of x-axis in table 1 (and other tables) should be training time instead of epochs, as we see in table 1, that algorithm's performance increases with longer epoch training. For example assuming DINO + HSV is twice slower than DINO, it would be great to see a performance comparison between DINO+HSV at epoch 100 with DINO at epoch 200 (similarity with other baselines).

3. It might be good to see some qualitative results on how good the learned features are with HSV strategy. For example, adding some attention maps as done in DINO paper, to get an idea if the attention maps also improves using HVS.

4. As HSV is proposed as a general view sampling strategy it might be helpful to see some results on other relevant baselines like SimCLR_v2, DINO_v2, Swav. If some of the recent methods require huge amounts of compute or dataset, showing some results on ImageNet with lower training epochs would give the reader a better idea about the efficacy of the hard view selection strategy.

5. It could be a good idea to relate and mention in the sub-section of “Taking a Closer Look at the Intersection over Union” with the idea of “local-to-global” correspondences (as introduced in SimCLR :  Global and local views/Adjacent views and DINO : global and student view). Qualitatively, looking at Fig2, it does look like a HSV selected view pair comprises of global view (larger area crop) and student view (small area crop).

**Questions:**

Some questions/suggestions:

1. How many seeds are used for Table 1 average computation. “Ref: averaged over multiple seeds unless otherwise mentioned”

2. It is stated in the paper that SimSiam+HSV has 1.55x overhead and DINO+HSV has 2x overhead. Since DINO uses much more views (10 views leading to 128 max view pairs for forward call) than SimSiam (2 views that leads to 4 view pairs for forward call). What is the time taken for doing 1 DINO forward call and 1 backward call as the reviewer was expecting the DINO+HSV to be slower than 2x.

3. The reviewer was wondering about the combination of HSV with random view pair selection in order to reduce the computational  overhead. Maybe having 50% random and 50% HSV samples for training would be nice ablation experiment to shed light on the requirement of HSV adversarial pair selection. If this works without decreasing the performance with a large margin, then it would be interesting to see some ablation on tuning this percentage of HSV samples in training.

---

> ### Author Response · Authors · 2023-11-12
>
> **[...] Ablation experiments regarding the initial number of view pairs that determine the hardness of the selected view pair is also provided.(higher initial pool of view pairs would lead to more adversarial view selection for CL training). The paper is well-written and easy to follow. The paper provides code for reproducibility.**
>
> We thank the reviewer for their constructive feedback and for acknowledging our efforts to make our work transparent and reproducible.
>
>
> **A detailed analysis of the HVS overhead in terms of running time/ wall clock/ FLOPs/ throughput along with memory requirement for each the baselines DINO (2x overhead), SimSiam(1.55x overhead), SimCLR. Preferably in a format of a table. [...] Extending on the above point, it seems since the overhead is added due to HSV, a more fair choice of x-axis in table 1 (and other tables) should be training time instead of epochs, as we see in table 1, that algorithm's performance increases with longer epoch training. For example assuming DINO + HSV is twice slower than DINO, it would be great to see a performance comparison between DINO+HSV at epoch 100 with DINO at epoch 200 (similarity with other baselines).**
>
> We have addressed the concern on computation overhead in the common comment to all reviewers. Please let us know if your concerns persist so that we can further discuss them with you.
>
>
> **It might be good to see some qualitative results on how good the learned features are with HSV strategy. For example, adding some attention maps as done in DINO paper, to get an idea if the attention maps also improves using HVS.**
>
> This is an interesting idea, which indeed may lead to insights as to what type of features HVS learns. We are working on this and will report back to the reviewer here once we have these figures.
>
> **As HSV is proposed as a general view sampling strategy it might be helpful to see some results on other relevant baselines like SimCLR_v2, DINO_v2, Swav. If some of the recent methods require huge amounts of compute or dataset, showing some results on ImageNet with lower training epochs would give the reader a better idea about the efficacy of the hard view selection strategy.**
>
> We thought carefully and long about the selection of baselines for our paper. Indeed, there exist other baselines as well. For DINOv2, we kindly note that it was not pretrained on ImageNet-1k but on the unreleased LVD-142M dataset and so there would be no way to add HVS and compare against it fairly. For SimCLRv2, we note that only a simplified version has been open-sourced by the authors [1]. For example, the public code of v2 does incorporate contrastive learning which is used in v1. In order for us to show that our method works with the original (contrastive) v1 loss, we decided against using v2. In the case of Swav, we notice that it comes from the same authors as DINO which is the more recent and better-performing method that, similar to Swav, also incorporates the multi-crop technique.
>
> Regarding the reviewer’s comment on including a more recent baseline, we hope we were able to address it by adding the popular iBOT baseline (see general comment to all reviewers above). We believe this result again underpins the general applicability of HVS since it also works well with a MIM objective.
>
> [1] https://github.com/google-research/simclr/issues/115
>
> **It could be a good idea to relate and mention in the sub-section of “Taking a Closer Look at the Intersection over Union” with the idea of “local-to-global” correspondences (as introduced in SimCLR [...]**
>
> We thank the reviewer once again for this valuable feedback point. Indeed, this is a rich addition to the existing description and builds a useful analogy to SimCLR as well as DINO through the local-to-global correspondences and multi-crop settings. We have added this to the end of the mentioned subsection in the paper.
>
> **How many seeds are used for Table 1 average computation. “Ref: averaged over multiple seeds unless otherwise mentioned”**
>
> Thank you for pointing this out. Our results are averages of 3 seeds (except for iBOT where we currently average over 2 seeds). We have updated the mentioned sentence in the paper to be more specific.
>
> **What is the time taken for doing 1 DINO forward call and 1 backward call as the reviewer was expecting the DINO+HSV to be slower than 2x.**
>
> Did we answer your question with the common comment to all reviewers and with the additional method-specific speed information provided there? Please do let us know if your concerns persist so that we can provide the missing information.

---

> > ### Author Response · Authors · 2023-11-12
> >
> > **The reviewer was wondering about the combination of HSV with random view pair selection in order to reduce the computational overhead. Maybe having 50% random and 50% HSV samples for training would be nice ablation experiment to shed light on the requirement of HSV adversarial pair selection.**
> >
> > We appreciate the suggestion of this interesting idea. In fact, we have tested this for the min IoU policy (see Table 4 in the paper; denoted as “alternating”) where it helped reduce the inferior performance difference to the baseline. Indeed, applying an alternating strategy to HVS would help with the speed overhead and likely reduce the computation overhead by approx. 50%. However, we hypothesize it would also reduce the final performance noticeably. In any case, this is an interesting idea to test and which we have also added to the list of actionable steps (see common comment ) to take for reducing the computational overhead
> >
> > Overall, we thank the reviewer again for the valuable feedback. Hopeful that we have addressed and mitigated your initial concerns, we would like to ask you if you could increase your score (or, alternatively, we would be glad for any points we should address that would raise your score).

---

> ### Author Response · Authors · 2023-11-14
> **Attention Map Results**
>
> Dear reviewer,
> We wanted to follow up on your suggestion to create attention maps as a way to qualitatively study the potential effects and differences of the HVS learning method and their resulting features when compared to the baseline. We now added an overview of the generated attention maps under Section “J Attention Maps” in our supplementary material (last 3 pages), where we contrast 9 attention maps from a DINO ViT-S/16 100-epoch model for each HVS and the baseline.
>
> All input images are from the ImageNet-1k validation split. The color code used depicts strong attention in yellow and weak attention in green and attentions from the HVS model are shown in the top row and attentions from the baseline in the bottom row, respectively. To summarize our take-away from this study, we do not see apparent strong differences between HVS and the baseline. What we occasionally notice is that the HVS models seem to capture the shape of some subtle/indistinct objects better (e.g. the creek/river in the valley in Fig. 13, the lizard in Fig. 14, the speaker in Fig. 15) or focus slightly more on the context and background (Fig. 17, Fig. 18, and Fig. 19). However, we note that these characteristics also often seem subtle. Would the reviewer agree?

---

> > ### Comment · Reviewer_PWAF · 2023-11-22
> > **Thank you for the response**
> >
> > Thank you for providing detailed explanation for the questions. It is encouraging to see the improvement after "normalizing Table 1 for wallclock time and applying the 50% resolution reduction" and it would helpful for the readers to add this "normalized table1" result to the main paper (or appendix by adding a reference for this table in main paper). Accordingly, I have increased my score to 6.
> >
> > Yeah I agree with the authors that the attention maps does look quite similar (with or without HVS).

---

### Official Review · Reviewer_seyJ · 2023-11-01

**Soundness:** 2 fair
**Presentation:** 2 fair
**Contribution:** 1 poor
**Rating:** 3
**Confidence:** 4

**Summary:**

In pair-based self-supervised representation learning methods (e.g. SimCLR), positive pairs are constructed using data augmentation.
This paper proposes to produce many random positive pairs (e.g. 4) for any query image and compute loss over the "hardest" one.
In a way, it is a hard positive (or view) selection method.
It is shown that this technique can be utilized for various methods such as DINO, SimSiam and SimCLR, yielding consistent improvements on ImageNet classification and several transfer learning scenarios.

**Strengths:**

The main idea behind the paper is to make the training of pair-based self-supervised methods harder.
This is done by producing several random positive pairs and backpropagating gradients through the hardest one.
It is shown that this strategy picks pairs that overlap less, hence models learn better representations on ImageNet-1K after being trained the same amount of "epochs".
Better representations mean consistent improvements on ImageNet-1K classification and various transfer learning experiments including classifcation, detection and segmentation.

Overall, the paper is easy to read.

**Weaknesses:**

I have two main concerns, listed below.

1) As shown in Figure-3, positive pairs used for computing loss overlap less thanks to the proposed method, and this facilitates better representations. Tian 2020b already shows this phenomenon, i.e. there is a sweat spot in the IoU ratio which leads to the "optimal" performance. It is not clear what more this paper offers. One distinct angle in this paper is the fact that multiple positive pairs are utilized during training (although loss is computed over only 1 pair in the end). I wonder what would happen if loss was computed over all crops, by simple averaging or weighted averaging (depending on the difficulty of a pair).

2) The computational overhead introduced by the proposed method ("about a factor of 1.55× for SimSiam") is a bit unfair for the baselines. I wonder if this extra compute time can be used in favor of other models too, e.g. by training models longer. Cause longer training schedules often bring substantial gains for self-supervised methods. Also, it should be noted that the proposed method has seen more samples (due to encoding multiple pairs) which already impacted for instance batch-norm statistics (although loss is not backpropagated over unused pairs). Then it would be nice to see baselines processed 4x more samples.

**Questions:**

I would like the authors to address the concerns I raised in the weaknesses part.

---

> ### Author Response · Authors · 2023-11-12
>
> **As shown in Figure-3, positive pairs used for computing loss overlap less thanks to the proposed method, and this facilitates better representations. Tian 2020b already shows this phenomenon, i.e. there is a sweat spot in the IoU ratio which leads to the "optimal" performance. It is not clear what more this paper offers.**
>
> We thank the reviewer for pointing out the relevance of Tian2020b. Indeed, we agree with the reviewer that in hindsight we should have contrasted our paper in greater detail with Tian2020b and regret to not have done that in the initial version of the paper. Since also Reviewer #a1az has brought up this point, we are copy-pasting our response here, which we believe discusses the similarities and differences to Tian2020b comprehensively:
>
> First, Tian2020b leverages the information of views to characterize good views for a given task based on the mutual information principle and use this framework to discover sweet spots for data augmentation hyperparameters. These data augmentation hyperparameters are widely adopted by SSL approaches and our baselines. Additionally, Tian2020b use their insights about good views to then adversarially learn flow-based models that generate novel color spaces for the small STL-10 dataset. After the views have been learned offline, they then perform standard contrastive learning on these generated views.
>
> This approach is different to HVS on multiple fronts: with HVS we propose a method that can be integrated *online* into CL and non-CL pipelines without the use of a potentially instable adversarial learning objective. Our method is dependent on the model state and on individual samples since it is integrated online into the learning procedure, which is different to Tian2020b since their approach is not entangled with the model state.
>
> While Tian2020b ablates also based on distance metrics in their study, they do not include the IoU metric. Moreover, another central difference is that we use the distance metric as a tool to better understand the selection of HVS and to discover that model-state and image-dependence is essential, where Tian2020b uses it as a motivation for their framework. Furthermore, our ablation studies include experiments that quantify the difficulty of predicting good/hard pairs, which Tian2020b does not have. In summary, we view Tian2020b as complementary to our work, particularly in the context of their derived data augmentation policies and believe that HVS could even be combined with the findings of Tian2020b. However, it's important to underscore the limited overlap in methodology and experimental design between the two approaches. HVS in the end could be seen as an early work that shows that for good views, we need to derive better sampling methods that can dynamically adapt to the actual state of training.
>
> We have now added the expanded discussion on Tian2020b from above to the Introduction and Related Work sections of our paper.
>
> **[...] One distinct angle in this paper is the fact that multiple positive pairs are utilized during training (although loss is computed over only 1 pair in the end). I wonder what would happen if loss was computed over all crops, by simple averaging or weighted averaging (depending on the difficulty of a pair).**
>
> Thank you for this idea on averaging the losses. Please allow us to make sure that the method was correctly understood by the reviewer. We emphasize that the loss is computed for all pairs individually but only backpropagated for one pair (namely the one with the highest loss). Let us make sure we understood your suggestion correctly: instead of forwarding all view pairs and discarding all except the pair with the highest loss, you are suggesting to keep all pair losses and compute an average across all pair losses or smart averaging (by weighing the pairs with some difficulty score)? Assuming we understood your suggestion, we believe it would significantly ramp up the computation cost of the method since it would effectively multiply the batch size (since all activations for all pairs would have to be kept in memory for the backward pass). While applying HVS with a weighted loss that amplifies the gradient updates proportional to some difficulty measure sounds very exciting, but since we have no experience in this regard we lack the intuition about the consequences of such a modification with regard to training stability. Please let us know if we misunderstood your suggestion so that we can engage in a follow-up discussion.

---

> ### Author Response · Authors · 2023-11-12
>
> **The computational overhead introduced by the proposed method ("about a factor of 1.55× for SimSiam") is a bit unfair for the baselines. I wonder if this extra compute time can be used in favor of other models too, e.g. by training models longer. Cause longer training schedules often bring substantial gains for self-supervised methods. Also, it should be noted that the proposed method has seen more samples (due to encoding multiple pairs) which already impacted for instance batch-norm statistics (although loss is not backpropagated over unused pairs). Then it would be nice to see baselines processed 4x more samples.**
>
> Thank you for communicating this critique with us. We have addressed the concern on computation overhead in the common comment to all reviewers. Please let us know if your concerns persist so that we can further discuss them with you.
>
> Regarding the concern about the models having seen more samples, we kindly disagree with the reviewer. First, the DINO and iBOT ViT-S models do not use batch norm but layer norm layers which do not keep track of running statistics and those experiments still show the same improvements. Moreover, even when batch norm layers are used, the batch norm learnable parameters are not affected by the HVS forward passes since we disable gradients for the selection step and hence, the BN parameters are not updated during the selection. Note also, that the running mean and variance buffers are only used for inference, see the pytorch documentation [1] which states:
>
> “*Also by default, during training this layer keeps running estimates of its computed mean and variance, which are then used for normalization during evaluation.*”
>
> During training, we keep our model in model.train() state so the running mean and variance are not used. Consequently, the HVS models do not have an advantage over the baselines in terms of having seen more samples.
>
> [1] https://pytorch.org/docs/master/generated/torch.nn.BatchNorm2d.html?highlight=batchnorm2d#torch.nn.BatchNorm2d
>
>
> We express our gratitude once more for the constructive feedback provided. Aiming to have effectively addressed your concerns, we kindly inquire whether there is an opportunity to increase your score. Alternatively, we welcome any additional points for improvement that would allow for a score increase.

---

### Official Review · Reviewer_a1az · 2023-11-06

**Soundness:** 3 good
**Presentation:** 2 fair
**Contribution:** 2 fair
**Rating:** 5
**Confidence:** 5

**Summary:**

This paper introduces a novel approach called Hard View Selection (HVS) to enhance self-supervised methods by improving the selection of image views during training. HVS effectively increases the task difficulty during pretraining and achieves competitive or better results compared to the conventional baseline, improving accuracy on ImageNet and transfer tasks across various methods.

**Strengths:**

- The method is simple to understand and sound
- Augmentations are crucial in self-supervised learning. However, they are either hardcoded or grid-searched. It is good to see that there is research in the direction of picking augmentations automatically
- The authors try many methods and validate with many downstream tasks

**Weaknesses:**

- The title says “contrastive” but most of the experiments use non-contrastive methods like simsiam and dino. This is confusing.
- In addition to the point above, it actually seems that the proposed method is better defined for non-contrastive losses. In fact, for contrastive losses (which contrast the attraction of the positives with the repulsion of the negatives) the method is not guaranteed to find the hardest possible set of positives and negatives. This would require exploring all the possible combinations of negatives, which is intractable. A better idea would be to use all the negatives from all the views at the denominator. In that case, since the denominator works as a max at low temperatures, you would be “guaranteed” to have the hardest negative. This is not described and discussed in the paper except one sentence at the end of 3.3.
- In algorithm 1, the authors report that they first organize the views into pairs and then forward each pair. This entails forwarding every view multiple times (one view can be in more than one pair). This is confusing and does not make sense at all. It would be better to first forward the views separately and then compare them 2 by 2. Can the authors please explain why they first create the pairs and then forward?
- Why not just make all comparisons and backprop all of them? Why is this not ablated?
- Figure 4 shows that the manual min IoU policy works for SimSiam but not for DINO. Maybe it’s just that multi-crop does not play well with min IoU. I am not fully convinced by this ablation.
- Figure 5 shows that performance decreases over time, which might be due to overfitting (or might not). In any case, cifar100 is not a good benchmark to make a point about longer training.
- Throughout all the tables the performance improvements are small. Once factored in the ~2x slowdown, there seems to be almost no difference. This is not great, also considering that the proposed method probably uses a lot more memory than the vanilla method, and it is also more complex to implement. In practice, I would rather train longer with a well-known vanilla method than using the proposed method.
- In addition, there exist more efficient ways to learn representations nowadays (e.g. CLIP). There are many papers that use multiple views for CLIP as well. Why not testing them?
- There are many other papers that treat this topic. They were not discussed. A few examples:
There are many other papers that treat this topic. They were not compared with. A few examples: "On the Importance of Asymmetry for Siamese Representation Learning", "What Makes for Good Views for Contrastive Learning?".

**Questions:**

See weaknesses.

---

> ### Author Response · Authors · 2023-11-12
>
> **The title says “contrastive” but most of the experiments use non-contrastive methods like simsiam and dino. This is confusing.**
>
> Thank you for pointing this out. We agree with you that the title is indeed misleading and may suggest that HVS is applicable to only contrastive learning. We changed the title to “Hard View Selection for Self-Supervised Learning”. Do you agree that title is better suitable?
>
> **In addition to the point above, it actually seems that the proposed method is better defined for non-contrastive losses. In fact, for contrastive losses (which contrast the attraction of the positives with the repulsion of the negatives) the method is not guaranteed to find the hardest possible set of positives and negatives. This would require exploring all the possible combinations of negatives, which is intractable. A better idea would be to use all the negatives from all the views at the denominator. In that case, since the denominator works as a max at low temperatures, you would be “guaranteed” to have the hardest negative. This is not described and discussed in the paper except one sentence at the end of 3.3.**
>
> We agree with the reviewer that this point is described vaguely and requires more clarity in the paper. We describe this in greater detail in the Appendix under “I.1 SimCLR”. If we understood the reviewer right, yes, that is correct and we updated I.1 with the following description for more clarity:
>
> “*For each iteration, we evaluate all possible view pairs and contrast each view against every other example in the mini-batch. Intuitively, the pair that yields the highest loss is selected, which is the pair that at the same time minimizes the numerator and maximizes the denominator in the above equation. In other words, the hardest pair is the one, that has the lowest similarity with another augmented view of itself and the lowest dissimilarity with all other examples.*”
>
> Did this address your concern and did we understand you correctly? We hope we were able to give some clarification and are happy to discuss further suggestions.
>
> **In algorithm 1, the authors report that they first organize the views into pairs and then forward each pair. This entails forwarding every view multiple times (one view can be in more than one pair). This is confusing and does not make sense at all. It would be better to first forward the views separately and then compare them 2 by 2. Can the authors please explain why they first create the pairs and then forward?**
>
> Thanks, you are right. In practice, we indeed only forward each view once to generate its embedding before we create the pairs over the embeddings. We fully agree with the reviewer that Algorithm 1 does not reflect this properly. For didactic reasons, we thought that it would be more understandable to a reader if the view pairs were first generated. However, after considering the reviewer’s feedback, we agree it would be better to stay closer to the actual (more efficient) implementation and updated Algorithm 1 accordingly. Thank you for pointing this out.
>
> **Why not just make all comparisons and backprop all of them? Why is this not ablated?**
>
> While we agree with the reviewer that the result of this experiment would certainly be interesting and valuable, its execution would require significantly larger compute resources because instead of selecting one pair from many and only computing the gradients for this pair, one would now have to compute and hold all gradients for all pairs which would drive up the memory and speed costs.

---

> > ### Author Response · Authors · 2023-11-12
> >
> > **Figure 4 shows that the manual min IoU policy works for SimSiam but not for DINO. Maybe it’s just that multi-crop does not play well with min IoU. I am not fully convinced by this ablation.**
> >
> > We thank the reviewer for this constructive feedback and would like to explain the ideas behind this mentioned ablation:
> >
> > Being aware of the computation expenses introduced by HVS, we thought about ways to simplify the method while maintaining its performance improvements. For this, we have adopted the idea of using distance metrics such as the IoU to incorporate into a simple but generally applicable learning strategy such as HVS is.
> >
> > As the reviewer pointed out rightly, the min IoU policy did not show the improvements across all baselines (specifically DINO) that HVS is able to provide. However, we emphasize that this policy was derived from running the SimSiam baseline. Since the main goal of this ablation was to be able to come up with a method that is generally applicable and transferable, we deemed it irrelevant to derive another policy with the DINO baseline. In our view, it is also much less straightforward to derive and apply such a policy in the multi-crop setting (let alone in other settings such as iBOT’s Masked Image Modelling objective) than it is to simply apply HVS. Instead, our interpretation of this ablation is that HVS’ model state and image-dependent selection of pairs is essential for the success of such a method. As a result, and agreeing with the reviewer, we believe that a manually derived min IoU policy (such as in the case of SimSiam) does not transfer to other baselines that entail variations in the augmentation pipeline (such as multi-crop).
> >
> > We hope we were able to convince the reviewer about the usefulness of this ablation and added the above explanation to the paper in Section “5.2 Q2: Can a Manual Augmentation Policy be Inferred?”.
> >
> > **Figure 5 shows that performance decreases over time, which might be due to overfitting (or might not). In any case, cifar100 is not a good benchmark to make a point about longer training.**
> >
> > We agree with the reviewer that CIFAR100 is not an ideal dataset for benchmarking longer training. Due to computational limitations mentioned, we were not able to conduct these longer trainings on ImageNet. For instance, the 300-epoch DINO model required 21 days only for pretraining. For Fig. 5 one would have to run multiple such trainings for many months. Nevertheless, we believe that this benchmark still yields interesting insights with regard to longer trainings: HVS is able to improve and maintain the increased performance difference, and hard views allow being less prone to overfitting when compared to the baseline. Would the reviewer agree? We are very open to engage in a discussion and suggestions to improve this experiment.
> >
> > **Throughout all the tables the performance improvements are small. Once factored in the ~2x slowdown, there seems to be almost no difference. This is not great, also considering that the proposed method probably uses a lot more memory than the vanilla method, and it is also more complex to implement. In practice, I would rather train longer with a well-known vanilla method than using the proposed method.**
> >
> > We thank the reviewer for these constructive points. Hoping that we were able to address the critique regarding time and memory complexity of HVS in the general comment to all reviewers above, we additionally emphasize that implementing HVS is very simple and we invite the reviewer to briefly take a look at our code. All it takes to integrate HVS is to change the dataloader to sample “num_views” images [1] (in contrastive/non-contrastive learning one does this anyway since we always need two views at a minimum) and integrate our simple select function that is called in the train loop [2]. We are currently preparing the integration of HVS into pytorch through a pull-request where HVS could be integrated as a decorator for the train function and further ease the adoption of it.
> >
> > We point out again that so far we have only seen evidence pointing to the fact that longer trainings with HVS also continue to improve the performance so that one can expect that the performance difference to the baseline is maintained.
> >
> > [1] https://anonymous.4open.science/r/hard-view-selection/data.py (Line 31)
> >
> > [2] https://anonymous.4open.science/r/hard-view-selection/pretrain.py (Line 196)

---

> > > ### Author Response · Authors · 2023-11-12
> > >
> > > **In addition, there exist more efficient ways to learn representations nowadays (e.g. CLIP). There are many papers that use multiple views for CLIP as well. Why not testing them?**
> > >
> > > Thank you for pointing this out. While we agree with the reviewer that other, multi-modal approaches exist to learn representations in a self-supervised way, we are unsure about the claim that those are necessarily more efficient.
> > >
> > > It's worth noting that our current focus in this work has been on vision-only approaches, and our choice of baselines reflects that emphasis. We kindly ask the reviewer to consider the achievements and contributions made in this study rather than dwelling on aspects from other areas such as multi-modal self-supervised learning. Regarding the potential adoption of HVS in multi-modal settings, we acknowledge its relevance and importance and would view it as a valuable avenue for future exploration. We hope the reviewer can appreciate the intriguing results we have presented in this work, and we sincerely value the constructive feedback provided.
> > >
> > > **There are many other papers that treat this topic. They were not discussed. A few examples: There are many other papers that treat this topic. They were not compared with. A few examples: "On the Importance of Asymmetry for Siamese Representation Learning", "What Makes for Good Views for Contrastive Learning?".**
> > >
> > > We thank the reviewer for pointing out our shortcoming on referencing relevant related work. While we agree with the reviewer that we do not include the first mentioned paper, we do discuss Tian2020b (which is “What Makes for Good Views for Contrastive Learning”) in our paper. However, we agree with the reviewer that our paper would benefit from a deeper discussion about the similarities and differences with Tian2020b. First, Tian2020b leverages the information of views to characterize good views for a given task based on the mutual information principle and use this framework to discover sweet spots for data augmentation hyperparameters. These data augmentation hyperparameters are widely adopted by SSL approaches and our baselines. Additionally, Tian2020b use their insights about good views to then adversarially learn flow-based models that generate novel color spaces for the small STL-10 dataset. After the views have been learned offline, they then perform standard contrastive learning on these generated views.
> > >
> > > This approach is different to HVS on multiple fronts: with HVS we propose a method that can be integrated *online* into CL and non-CL pipelines without the use of a potentially instable adversarial learning objective. Our method is dependent on the model state and on individual samples since it is integrated online into the learning procedure, which is different to Tian2020b since their approach is not entangled with the model state. While Tian2020b ablates also based on distance metrics in their study, they do not include the IoU metric. Moreover, another central difference is that we use the distance metric as a tool to better understand the selection of HVS and to discover that model-state and image-dependence is essential, where Tian2020b uses it as a motivation for their framework. Furthermore, our ablation studies include experiments that quantify the difficulty of predicting good/hard pairs, which Tian2020b does not have.
> > >
> > > In summary, we view Tian2020b as complementary to our work, particularly in the context of their derived data augmentation policies and believe that HVS could even be combined with the findings of Tian2020b. However, it's important to underscore the limited overlap in methodology and experimental design between the two approaches. HVS in the end could be seen as an early work that shows that for good (or hard) views, we need to derive better sampling methods that can dynamically adapt to the actual state of training.
> > >
> > > Regarding the first mentioned paper, “On the Importance of Asymmetry for Siamese Representation Learning”, we are doubtful about the relatedness to our work after reviewing it. The focus of this paper appears to be on improving representation learning by controlling encoding variance in the latent space of Siamese networks, exploring their findings with different augmentation strategies such as multi-crop. While we acknowledge the potential for combining HVS with their approach, the immediate connection to our work may require further clarification. If the reviewer identifies relevant points of intersection between our work and the mentioned paper, we are open to incorporating those insights.
> > >
> > > We have now added the expanded discussion on Tian2020b from above to the Introduction and Related Work sections of our paper. We thank the reviewer once again for the valuable feedback. Hopeful that we have addressed and mitigated the initial concerns, we would like to ask you if you could increase your score (or, alternatively, we would be glad for any points we should address that would raise your score).

---

### Author Response · Authors · 2023-11-12
**Common Comment to All Reviewers (1/2)**

We are very grateful for the helpful feedback and would like to thank the reviewers for their time and effort and are looking forward to a lively discussion. Before focusing on each of the reviewer’s concerns individually, we would like to communicate two main points in this general comment section addressed to all reviewers:

* Application of HVS to an additional baseline and results
* Concern about HVS time overhead

**Application of HVS to an additional baseline**
In preparation for the rebuttal, we applied HVS to a fourth, well-known baseline “iBOT” [1] which, in essence, combines DINO with a Masked Image Modelling (MIM) objective. We are happy to report that for a 100-epoch pretraining, iBOT+HVS (on ViT-S/16) achieves 70.41% linear eval. accuracy which is an **improvement of +0.91%*** over the iBOT baseline (again self-reproduced; all values computed across 2 seeds for each baseline and HVS). For the 300-epoch runs, the **improvement is even larger with +1.51%**.  We have added these results to Table 1 in the paper. They again underpin the general applicability of HVS since it also works well with a MIM objective.

(Update Nov 18: we have added the 300-epoch performance results and updated the 100-epoch results which are all now based on two seeds.)

**Concern about HVS overhead in terms of running time**
Some reviewers have raised a concern about the computational overhead that HVS introduces. This is a valid concern and we want to improve the paper’s transparency further. Reviewer #PWAF suggested incorporating a table listing the computational overhead and we have added such a table to Appendix H, along with the following discussion. The computational overhead factors compared to baseline are as follows: SimCLR (x1.69), SimSiam (x1.55), DINO (x2.15) and iBOT (x1.50). For DINO’s 2 global and 8 local views (default), applying HVS with nviews=2 sampled for each original view results in 4 global and 16 local views. Since considering all combinations would yield over 77k unique comparisons (4 over 2 times 16 over 8), to remain tractable, we limit the number of total comparisons to 128.

While technically there can be a memory overhead with HVS, with the number of sampled views chosen in this paper, the backward pass of the methods that compute gradients only for the selected view pair still consumes more memory than the selection part of HVS (even for 8 sampled views in SimSiam!). Note, that selection and the backward computation are never executed at the same time but sequentially.

Our central point here is to emphasize that the time overhead factors were measured without any optimization of HVS’ efficiency and, in our view, there are multiple ways to improve it. In our work, we opted for the easiest implementation possible to showcase that selecting harder views dependent on the model learning state can help boost performance in contrastive learning. Going forward with more compute-efficient HVS solutions, one could think of:
* using smaller resolution views for the view selection, as done in the multi-crop [2] method used in DINO or (*)
* using embeddings of views from “earlier” layers in the networks or
* using 4/8 bit low-precision for the view selection or
* using one GPU just for creating embeddings and selecting the hardest views while the remaining GPUs are used for learning or
* switching between HVS and the standard pipepline in alternating fashion or
* bypassing forwarding of similar pairs and more

(*) After receiving the reviewer feedback, we tested briefly for DINO ViT-S/16 and SimSiam ResNet-50 the effect of halving the image resolutions:
* DINO ViT-S/16: **~27% speed improvement (0.52 versus 0.38s per batch)** with global/local resolutions of 112² (instead of 224²) and 48² (instead of 96²)
* Simsiam RN50: **~20% speed improvement (0.20 versus 0.16s per batch)** with resolution 112² (instead of 224²)

(Note: We have just updated the speed test values after running another longer test with turned off background processes.)
This suggests that, with improvements like these, the speed costs of the HVS pipeline can be reduced.

Lastly, we want to add that, when multi-crop learning was introduced in [2] and used extensively in DINO, the computational slowdowns of 25% (see SSL cook book [3] Section 3.1.1) were overlooked due to its impressive performance. However, our study operates under limited computational capacity as described in the paper which prevents us from surpassing SOTA models and demonstrating comparable performance. This underscores the prior disregard for slowdowns when ample resources were available.

---

> ### Author Response · Authors · 2023-11-12
> **Common Comment to All Reviewers (2/2)**
>
> Our central arguments can be summarized as follows:
> 1. There are little to no diminishing returns when training longer with HVS (seen for 300 epoch trainings on ImageNet and for 4000 epochs on CIFAR100 in Fig. 5).
> 2. When normalizing Table 1 with respect to training time, HVS still yields slightly better performances, even *without* any of the many possible efficiency improvements.
>
>
> Given these arguments, we see our current investigation as an early-stage exploration that highlights the novelty and efficacy of selecting more challenging views based on the model’s learning state for improving contrastive and non-contrastive learning performance. Our work may lay the groundwork for future explorations that could devise more efficient sampling methods to generate hard views and which in turn could benefit various SSL approaches.
>
> We have revised the PDF to integrate the valuable feedback from the reviews, enhancing the clarity of our points (with ongoing refinements as the rebuttal progresses). To facilitate a quick turn-around, we have not fine-tuned the main paper to still fit into 9 pages during this discussion period, but would of course do so upon acceptance.
>
>
> [1] https://arxiv.org/abs/2111.07832
>
> [2] https://arxiv.org/abs/2006.09882
>
> [3] https://arxiv.org/abs/2304.12210

---

### Author Response · Authors · 2023-11-20
**New results: reduced resolutions speed up HVS and retain performance**

Dear reviewers,

we are happy to report that we were successful in **speeding up the HVS method significantly by reducing the image resolutions for HVS’ pair selection while retaining the exact final performance**. We have tested this for DINO ViT-S/16 100-epoch pretraining with a 50% reduction of the original image resolution. As a consequence, the pretraining ran ~30% faster and we were able to reproduce the exact results from Table 1 across two seeds. We also measured the efficiency improvements for different resolutions on SimSiam and DINO:

| Resolution 	| SimSiam (RN50) 	| DINO  (ViT-S/16)	|
| -------- | ------- | ------- |
| 100% | x1.55 | x2.15 	|
| 75% 	| x1.45 	| x1.68 	|
| 50% 	| x1.24 	| x1.56 	|
| 25% 	| x1.16 	| x1.5 	|

When normalizing Table 1 for wallclock time and applying the 50% resolution reduction, we get the following improvements (100-epoch pretraining):

| Method 	| Architecture 	| Linear Eval. Acc.	|
| -------- | ------- | ------- |
| DINO | ViT-S/16 | 74.06 	|
| DINO + HVS	| ViT-S/16 	| 74.67 	|
| **Improvement**	|  	| **+0.61** 	|
| SimSiam | RN50 | 68.46 	|
| SimSiam + HVS	| RN50 	| 68.98 	|
| **Improvement**	|  	| **+0.52** 	|
| SimCLR | RN50 | 64.31 	|
| SimCLR + HVS	| RN50 	| 65.33 |
| **Improvement**	|  	| **+1.02** 	|

We are still working on adding the iBOT ViT result to this comparison as well. Once all results are calculated, we will integrate the normalized table in the final version of the paper.

All in all, we believe these results effectively address the majority of the reviewer’s concerns regarding the efficiency of HVS. We would greatly appreciate it if the reviewers could kindly consider this new information when evaluating our work and potentially revisiting their assessments.

Furthermore, we would like to note that unfortunately we have not received any feedback or engagement from the reviewers so far, and with the reviewer discussion period ending in just two days, we are eager to ensure that these insights are taken into account in the final evaluation of our work.

---

### Meta-Review · Area_Chair_1z5o · 2023-12-11

**Metareview:**

The paper proposes hard negative selection for contrastive learning, and demonstrates some potential gains. While the approach is relatively simple and interesting, achieving meaningful performance gains. The reviewers express several concerns, among which the significant one is the computation overhead of the proposed method over the baselines, sometimes even 1.5x or 2x more expensive than baselines, which makes it not a fair comparison. In author responses, authors provide a new method of using smaller image size for achieving similar performance, though the results were not complete. The authors can also compare with baseline with matching compute. The current paper draft, without the revision, is below the bar. I would encourage the authors to revise the paper integrating new results presented in the rebuttal as well as addressing other reviewers' concerns.

**Justification For Why Not Higher Score:**

Main concern on the computation cost and fair comparison is not fully addressed.

**Justification For Why Not Lower Score:**

N/A

---

### Decision · Program_Chairs · 2024-01-16

Reject